# HESS Opinions: The Myth of Groundwater Sustainability in Asia

Franklin W. Schwartz[1], Ganming Liu[2] and Zhongbo Yu[3]

[1]School of Earth Sciences, The Ohio State University, Columbus, OH, 43210, USA
[2]School of Earth, Environment and Society, Bowling Green State University, Bowling Green, OH 43403, USA
[3]State Key Laboratory of Hydrology-Water Resources and Hydraulic Engineering, Nanjing, 210098, China

*Correspondence to*: Franklin W. Schwartz (schwartz.11@osu.edu)

**Abstract.** Across the arid regions of water-stressed countries of Asia, groundwater production for irrigated agriculture has led to water level declines that continue to worsen. For India, China, Pakistan, Iran and others, it is unrealistic to expect

groundwater sustainability in a verifiable sense to emerge. Fragmented governance and the general inability to bring traditional socio-economic tools to bear on reducing groundwater demands have impeded progress to groundwater sustainability. For India and Pakistan, where operational management is at the level of states and provinces, there is no capacity to regulate. Also in both China and India, the tremendous numbers of groundwater users, large and small, confound regulation of groundwater. With business as usual, groundwater-related problems receive insufficient attention, a situation referred to as an "accelerating

and invisible groundwater crisis" (Biswas et al., 2017). Another obstacle to sustainability comes from trying to manage something you do not understand. With sustainable management, there are significant burdens in needed technical knowhow, in collecting necessary data, and in funding advanced technologies. Thus, there are risks that that Iran, India and Pakistan will run short of groundwater from over-pumping in some places, and also be adversely affected by global climate change.

## 1 Introduction

About 20 years ago, hydrogeologists began more fully to appreciate the extent of non-sustainable withdrawals of groundwater worldwide. However, recognizing a problem and doing something about it are two different things. Our focus here is Asia, where the need for sustainable groundwater management is essential, given impacts from irrigated agriculture and growing urbanization. Yet, progress to sustainable development has been slow to non-existent. For many of these countries and even others outside of Asia, groundwater sustainability is essentially just a myth. This paper makes a case that groundwater impacts

in developing Asian countries are already bad and getting worse. Further, it describes governmental and socio-economic impediments to sustainable groundwater management, along with evident deficiencies in available data and for most, absent capacity to either regulate or undertake necessary projects. There is a huge gap between the full-blown blown water management approaches with demonstrated compliance, for example, in Singapore, and California, USA and those in Asia.

The concept of sustainability refers to the "development and use of groundwater in a manner that can be maintained for an

indefinite time without causing unacceptable, environmental, economic or social consequences" (Alley et al., 1999). It builds

on the foundational concept of safe yield as "the limit to the quantity of groundwater which can be withdrawn regularly and permanently without dangerous depletion of the storage reserve" (Lee, 1915). A modern concept of groundwater sustainability recognizes the additional complexity provided by the inherent coupling of groundwater and surface water systems (Winter et al., 1998; Sophocleous, 1997) and a groundwater supply that is impaired because of contamination.

Progress toward aquifer sustainability also requires governments to provide a framework for action. The first step is developing a vision for groundwater management, which leads to policies and laws (Smith et al., 2016). The laws provide the authorities and tools needed to protect groundwater resources. Examples include explicit powers to locate and register wells, and tools to manage quantities pumped through licenses or "obligations" (Smith et al., 2016). They also could extend to (i) banning new-well construction in critical areas; (ii) "capping" withdrawal rates with existing wells (Garduno and Foster, 2010); and (iii)

prosecutions for illegal pumping or waste discharges (Smith et al., 2016). Legislation might also provide strict controls on the use and disposal of hazardous chemicals, or bans on the storage or use of hazardous chemicals in areas critical for groundwater. Laws and tools are ineffectual without an operational framework for managing groundwater in the field. Oversight is required at national levels, together with other jurisdictions that make sense, for example, states, river basins, aquifers, or local communities (Smith et al., 2016). Governments also are responsible for funding various programs, and coordinating activities

with implications for groundwater in areas of surface water, agriculture, energy etc. It is also important to anticipate unintended consequences from actions taken in other parts of the government, a theme we will return to.

Successful implementation of management programs depends on the support and willing participation of local stakeholders. When stakeholder views are represented, especially during the drafting of legislation and implementation, there is broader compliance with regulations, a willingness to collect and share basic groundwater data, and active self-regulation (Smith et al.,

2016; Garduno and Foster, 2010).

Governments can use socioeconomic tools to promote groundwater sustainability. For example, charging for the water will reduce its use, especially with large commercial users (Garduno and Foster, 2010). Of course, this step requires the difficult task of metering of water used in irrigation. A second example is strategic, governmental investment to provide for efficient irrigation. Another beneficial direction is reversing common macro-policies that actually promote groundwater over-use

(Garduno and Foster, 2010). Examples in India include price support for crops that need lots of water, and reduced or no costs for agricultural necessities, such as rural electric power and agrochemicals.

Had decision-makers acted on the basis of scientific knowledge, Asia, would be far along to groundwater sustainability. Earlier studies pointed out problems and posited solutions. For example, in 2000, the World Bank formed the Groundwater Management Advisory Team (GW-MATE). Over the next decade, team members worked in Asia and elsewhere on

groundwater-related issues and sustainable groundwater use. Their reports identified seriously impacted groundwater systems and provided practical approaches to sustainability. Yet, these ideas did not lead to substantive action because there was little in the way of operational frameworks and capacity for management.

**2 Trends in Depletion and Contamination of Groundwater Continue to Worsen**

In China, India, Pakistan and other hotspots (Figure 1), the impacts to groundwater due depletion and contamination are continuing to worsen for reasons that we will discuss in Section 3. China's most visible groundwater problem is associated with the over-production of groundwater from aquifers underlying the core of the North China Plain (Figure 2a). Groundwater withdrawals since 1960 have produced excessive drawdowns that began to receive attention about 15 years ago. Water-level declines of > 20 m were evident in the shallow unconfined aquifer and > 40 m in the deep freshwater aquifer (Foster and

Garduno, 2004). Estimated reductions in groundwater storage were of the order of -8.8 km$^3$ yr$^{-1}$. By about 2010, drawdowns as high as ~60 m were reported in the unconfined aquifer (Cao et al., 2012) and >80 m in the deep freshwater aquifer (Zheng et al., 2010) (Figure 2a).

The aquifers of the North China Plains are essential to wheat production, to maintaining socio-economic contributions associated with the agricultural economy, and in supplying water to cities (e.g., Beijing and Baoding) (Foster and Garduno,

2012). However, continuing water-level declines indicate limited progress to sustainability. A recent assessment using GRACE (Feng et al., 2018) indicates almost constant declines in storage of -7.2 km$^3$ yr$^{-1}$ from 2002 to 2015. This result compares to an earlier estimate of -11.3 Gt yr$^{-1}$ (-11.3 km$^3$ yr$^{-1}$) (Rodell et al., 2018).These impacts will continue even with water available from the South to North Water Transfer (Ye et al., 2015; Bloomberg, 2017).

In China, there are other places with groundwater-related problems. Depletion is evident at some 164 locations, encompassing

¾ of China's provinces (Wang et al., 2018). Declining water levels have increased the size of the area affected by land subsidence – 4.9x10$^4$ km$^2$in the 1990s to 7.9x10$^4$ km$^2$ in 2000s to 9x10$^4$ km$^2$ in 2012 (Wang et al., 2018). Pumping of groundwater in coastal areas is also producing seawater intrusion.

The groundwater situation is also troubling in India with an annual production of ~250 km$^3$, the largest in the world. For India, groundwater provides 85% of drinking water and 60% of water for irrigation (World Bank, 2010). There are two prototypical

settings for groundwater in India. Shallow hard-rock aquifers, like the Deccan Traps (Basaltic Lava Flows) or weathered granitic rocks, occur across the upland areas of the Indian Peninsula (Figure 2b). These low yielding, weathered bedrock aquifers are important sources of water, which are being increasingly exploited with rates of withdrawal often greater than recharge (World Bank, 2010). Water typically occurs in fractures in the upper 25 m. During a typical year, increases in water levels due to recharge from monsoonal rains do not fully recover from the withdrawals of previous years.

The Indo-Gangetic alluvial (IGA) aquifer system occurs across the top of India (Figure 2b), extending into Pakistan, Nepal, and Bangladesh. It includes flood plains along the Indus and Ganges Rivers and their tributaries as a sequence of alluvial sediments > 200 m thick, derived from the Himalayan Mountains (MacDonald et al., 2016).

Groundwater production from the IGA aquifer system in 2010 (including adjacent countries) was 205 km$^3$ yr$^{-1}$, increasing at 2-5 km$^3$ yr$^{-1}$ due to continued expansion of irrigated agriculture (MacDonald et al., 2016). These large withdrawals are offset

by comparably large inflows as leakage from irrigation canals, irrigation return flows, and natural recharge from monsoonal rains. Assessments are complicated by spatial variability in hydraulic parameters, various water quality impacts, and

uncertainties in recharge estimates (MacDonald et al., 2016). Rates of storage depletion, estimated using GRACE data, range from 17.76 ±4.5 km$^3$ yr$^{-1}$ (Rodell et al., 2009) to 14 ±0.4 km$^3$ yr$^{-1}$ (Long et al., 2016). The most realistic estimate (2000-2012) is somewhat lower, 8.0 ±3.0 km$^3$ yr$^{-1}$ (5.2 ±1.9 km$^3$ yr$^{-1}$ for northern India), based on actual groundwater measurements (MacDonald et al., 2016).

Yet, impacts from pumping are not the urgent problem that some measurements (e.g., Rodell et al., 2009) imply. The relatively large quantities of groundwater stored in the upper 200 m of the IGA system coupled with 100 plus years of additional recharge from unintended canal leakage and irrigation return flows means that depletion is restricted to certain local areas. Yet, what is concerning is that the greatest recent water-level declines are evident in northern Indian and Pakistan, areas essential for food production with irrigation.

Often misunderstood, the threat to the sustainability of supplies from IGA aquifer system is associated with water quality issues – salinity, urban and industrial contaminants, and arsenic in groundwater (MacDonald et al., 2016; Foster et al., 2018; Young et al, 2019). The origin of salinity in the shallow groundwater is complex but commonly associated with effects of irrigation. Leaking canals, over more than a century in some instances, have led to waterlogging and salt accumulation in soil, and the salinization of recharge (Foster et al., 2018). Large capacity irrigation wells are also capable of mobilizing naturally salty water occurring at depth with up-coning. Estimates are that 18,000 km$^3$ or 60% of shallow groundwater in the IGA system suffers water-quality impairment (MacDonald et al., 2016).

The largest cities of India exemplify the emerging problems of water sustainability. A useful example is Delhi whose population of ~25 million is poised to double in the next 30 to 50 years. Most of Delhi's drinking water comes from surface-water sources; but groundwater from the IGA aquifer system is both important and problematic. Almost every sustainability issue just discussed is a major problem for Delhi – rapidly declining water levels, salinity at depth, and nitrate concentrations commonly >45 mg/L and as high as 1500 mg/L. Various news outlets have been active in expressing concerns about the local impacts of these problems (Text Box 1).

Political and policy failures associated with groundwater and surface water have created a crisis for India that bears directly on food, water and health (Biswas et al., 2017). With centuries of mismanagement of water resources, and "institutional incompetence" (Biswas et al., 2017) in the context of a large growing population, there has been no willingness for action politically in India beyond "cosmetic changes" (Biswas et al., 2017). Although issues involved with surface waters (contamination, fights over allocation, and reliability of public supplies) are worsening; "the groundwater situation is even worse" (Biswas et al., 2017). Yet, data on groundwater is poor in quality or unavailable. Rampant growth in groundwater utilization is linked in part to the failure of government to provide surface water for irrigation (Biswas et al., 2017).

Pakistan is another country with groundwater issues threatening future sustainability. Its large population, ~208 million and growing, contributes to its water scarce status with a per capita availability of water in the lowest 10% of the world's population (Young et al., 2019). The Indus River and its tributaries are significant surface-water resources, used almost entirely to support irrigated agriculture. Yet, the use of water is inefficient with significant losses due to canal leakage, evaporation, and over-irrigation (Young et al., 2019).

The IGA aquifer system extends southward into Pakistan along the length of the Indus River. For now, levels of groundwater in the IGA aquifer system in Pakistan are stable or even increasing (MacDonald et al., 2016). The main problems are associated with water-level declines of ~10 m since the 1980s in the important food growing area of Punjab Province to the northeast (Young et al., 2019). Here, as in India, canal leakage and irrigation return flow have continued to provide an unmanaged aquifer recharge system that has banked water in the subsurface since the late 1800s to the point of waterlogging in some places (MacDonald et al., 2016). The greater threat to sustainability often comes from the kinds of water quality problems mentioned previously.

There is little progress in the development of a sustainability ethic for groundwater management in Pakistan. Assessments are frustrated by an absence of data and the lack of a quantitative understanding of groundwater-surface water interactions along the major rivers (Young et al., 2019).

Elsewhere in Asia, the non-sustainable production of groundwater has resulted in even more serious problems. In Iran, the significant loss of groundwater resources could render major parts of the country uninhabitable with the possibility of millions displaced as conditions worsen (Collins, 2017). In addition to widespread declines in water levels, there are significant problems related to land subsidence and declining water quality (Madani et al., 2016).

Various factors have contributed to groundwater insecurity. Iran has a growing population of ~80 million, which has doubled over the last 40 years (Bozorgmehr, 2014). The country is dry, making groundwater a growing source for drinking and irrigation water. A continuing trend towards urbanization has resulted in an urban population of 70% with 18% in Tehran (Madani et al., 2016). Since 1999, there has been a succession of drought years. When coupled with an increase in annual temperature, the new normal is dryer and hotter weather with a likely decline in precipitation and recharge in coming decades due to climate change (Gohari et al., 2013; Nabavi, 2018).

This water crisis is also driven by socioeconomic decisions in the late 1970s to become self-sufficient in wheat, the country's most important crop (Collins, 2017). The expansion in wheat production through irrigation has had significant impacts on groundwater. Yet, there are few signs of movement to a more sustainable groundwater future (Collins, 2014).

Another Asian hot spot for impacts associated with unsustainable groundwater production is Jakarta, Indonesia, on the island of Java. Approximately 25-30% of the more affluent residents of this large city receive piped-in surface water (Colbran, 2009). Others obtain drinking water from large numbers of groundwater wells, rainwater, vendors, bottled water, etc. The poor quality of piped water has pushed industries and other large consumers to deep (~150 m) groundwater (Colbran, 2009).

Yet, this is not a drought story. Large, localized production from the shallow unconfined aquifer ~50 m thick and a deeper confined aquifer ~100 m thick is not sustainable even with significant natural recharge (Kagabu et al., 2013). The overuse of groundwater has been evident for a long time. For example, in 1995, reported pumping rates were three times larger than recharge rates. By 2008, drawdowns in the deep aquifer were >40 m with hydraulic heads 25 m below sea level (Kagabu et al., 2013). Water quality in the shallow aquifer is impacted by urban contaminants, like $NO_3$ (Kagabu et al., 2013) because there are virtually no sanitary sewer systems. There is evident seawater intrusion landward within the deep aquifer caused by over-pumping. Declining water levels have also resulted in subsidence that in several places exceeds 2 m (IRIDeS, 2017).

Now approximately 40% of the city's land surface is below sea level with only a seawall to protect land from inundation. Yet, there is no urgency around groundwater with business as usual.

## 3 What are the Hurdles to Groundwater Sustainability?

Developing Asian countries have encountered significant roadblocks hindering progress to groundwater sustainability. So far,
it has been relatively painless for countries and large cities simply to ignore groundwater issues, which in the case of India has been called an "invisible" crisis (Biswas et al., 2017). Of greater concern in Asian countries is a collection of more critical national issues related, for example, to growing their economies, feeding their people, maintaining national security, and improving the social conditions for growing populations.

In Asian countries, there much less deference to water security than food security. India's "Green Revolution" (GR) is a case
in point. In the 1950s, government leaders in India were troubled by the deaths from the Bengal Famine of 1943 (Rahman, 2015). With their growing population, achieving food security became a top priority. In the 1960s, the GR began with an expansion in agricultural lands, new high-yielding seeds, expanded irrigation, double cropping, and vastly increased fertilizer and pesticide applications (Rahman, 2015; Schmanski, 2008).  India became food secure with large increases in the production of food and cereal grains.

Yet there is dark side, which includes severe social, economic, and environmental problems, particularly in the amazingly productive Punjab region of India (Figure 2b). Examples include the high suicide rates of farmers, increasing cancer rates from pesticides, and especially the unsustainable use of groundwater (Schumanski, 2008; Singh and Park, 2018). Groundwater impacts were slow to develop but are now serious with total water-level declines ranging from 4.5 m to 35 m (Rahman, 2015). China, India, and Iran have been able to aggressively ramp up agricultural production to feed their people without adequately
considering the impacts to groundwater. With recharge and the inherent capacity of large aquifers to store abundant groundwater, problems developed incrementally and have been difficult to recognize in data-poor settings. Now, food production from irrigated agriculture is structurally part of the national economies of these countries, making it difficult to reduce the production of food and groundwater.

The second major impediment to sustainable management is limitations in terms of a socioeconomic framework for action and
necessary data. Following here is a discussion of these issues for three large countries India, Pakistan, and China.

As mentioned, centralized groundwater management requires appropriate policies, legislation, and a functional regulatory framework. India has a useful collection of laws in place but groundwater management is "weak to nonexistent" (World Bank, 2010). The most important limitation is that water is a state responsibility and many of the states in India have no capacity to monitor groundwater production or to enforce regulations, especially with many small water users. Thus, India's estimated 20
million wells are unregulated and outside any regulatory framework (World Bank, 2010).

Another drag on sustainability efforts in India is political sensitivities. For example, the free or nearly free electrical power for irrigation in rural areas is a significant factor in groundwater over-pumping. Yet, there is no political will to change this policy

(World Bank, 2010). It is also problematic that groundwater oversight exists in many government agencies with no clear definition of responsibilities between the state and central governments. Thus, for India, there are no realistic possibilities for state intervention in groundwater management (World Bank, 2010). The best opportunities exist at the community level.

The situation in Pakistan is quite similar to India with little prospect for progress. Within their federal system of governance, provinces have responsibilities for managing groundwater. However, there are neither provincial regulatory frameworks nor capacity to regulate access to groundwater (Young et al., 2019). There is limited technical capabilities in managing surface-water and groundwater conjunctively (Young et al., 2019). Thus, the long-term risk to groundwater sustainability in the Lower Indus River basin and delta from inefficient irrigation and seawater intrusion are unmitigated.

In China, there are legal/operational frameworks that have the potential to contribute to groundwater sustainability. A complex, multi-tiered system for water-resource administration exists, with units represented all the way down to local Water User Associations (Doczi et al., 2014). Most progress in the management of agricultural water has been with traditional surface-water irrigation systems, which has contributed to rises in production and irrigation efficiency (Doczi et al., 2014). Groundwater is more problematic with water-level declines continuing (Doczi et al., 2014; Biswas and Hartley, 2017; Wang et al., 2018). The number of privately owned wells has increased significantly giving farmers the ability to "protect their crops" (Doczi et al., 2014) from drought and the inconsistent availability of surface water for irrigation.

Doczi et al. (2014) point to problems in implementing national policies with regulation unable to catch up with the rapidly expanding use of groundwater. Specific measures like drilling permits, quotas on water, and fees have been implemented but neither broadly (Wang, et al., 2018) nor effectively (Doczi et al., 2014). Controlling water-level declines across the North China Plain is also complicated by the growing need for groundwater to help support rapid urban and industrial growth (Biswas and Hartley, 2017). However, China does have the fiscal and technical capacity to support projects focused on sustainability. The challenge in this respect is in finding water sources for MAR.

The Asian countries we have examined also have problems with data. Some water-level data are available in areas most impacted by over-pumping. However, information on what wells are pumping what quantities of water typically do not exist. It is evident that the tens-of-millions of wells in both China and India provide a formidable operational challenge in monitoring. With the pervasive irrigation associated with rivers and systems of canals, unmanaged aquifer recharge is a significant, yet unknown factor influencing water levels. There is even less information on threats to sustainability coming from groundwater contamination.

Development of some "understanding" in relation to the sustainable management of groundwater depends upon programs of hydrogeologic mapping, monitoring, and modeling. Yet, there is little discernable progress in data collection necessary to support sustainability initiatives in either India (Biswas et al., 2017) or Pakistan (Young et al., 2019). There may be somewhat more progress in China but information there is siloed and lacking in necessary transparency.

There are positive activities underway, which contribute to groundwater sustainability by promoting groundwater recharge and storage. Yet, just what that contribution actually is unknown in a technical sense. For example, India is the world leader in the number of installed systems to store water in the subsurface. (Dillon et al., 2019). There are several million traditional

(often old) recharge structures, such as, percolation tanks/ponds and streambed infiltration systems with millions more planned (Dillon et al., 2019). Yet, there are few quantitative assessments of how well these systems work in promoting recharge (Dillon et al., 2019; Dashora et al., 2018) and whether they can be effective without actions that manage demands and quantities pumped.

China, however, has a much smaller number of these kinds of traditional recharge projects in operation (Dillon et al., 2019). Instead, they are investing in their "sponge-city" concept, a collection of "low-impact" practices (Biswas and Hartley, 2017) designed primarily to reduce urban flooding and water pollution. The idea is to store water in the subsurface through pervious pavements or utilizing/storing storm water in rain gardens or rooftop gardens. These kinds of green infrastructure projects are growing in China. While contributing to groundwater sustainability through urban recharge, it is not yet clear what that contribution will be.

In the megacities, like Jakarta, Delhi, and Karachi, our reviews found the status of groundwater data to be meagre to nonexistent and inadequate to support technical or socioeconomic efforts to sustainability.

The kinds of technical knowledge and data needed for sustainable groundwater management are well known. They include a robust qualitative and quantitative understanding of how the land-based portion of the hydrologic system functions, physically, chemically and biologically. Basic data collection involves metering or other approaches to establish water-utilization, groundwater/surface-water interactions, aquifer characterizations, testing, sampling and measurements in the field, supported by various monitoring networks, data acquisition systems, laboratories and database systems. Figure 3 highlights the broad scope of data needs with an illustrative conceptual model of a complex coastal hydrologic system (CDWR, 2016).

Asian countries starting from scratch will need to anticipate costs associated with years of field operations in, for example, groundwater mapping, aquifer testing, and water quality measurements. Various monitoring networks will need to be designed and emplaced, as well as equipment to be purchased, installed, and operated. Provision must be made for data compilation and storage, interpretations, modeling, laboratory measurements, etc. What adds even more difficulty is an absolute need to monitor for one to several decades to provide an average set of baseline conditions (CDWR, 2016). The creation of conceptual models, water balance calculations, and compliance assurance all require these kinds of data. A useful place to gain perspective is with a series of best practices reports of the California Department of Water Resources (e.g., CDWR, 2016). They are intended to provide technical assistance for California's new state-wide initiative in sustainable groundwater management.

The third major obstacle is that technically oriented initiatives on sustainability require expensive infrastructure with continuing operating costs, especially with the development of new water sources for MAR, for example, treated sewage, rainwater, or imported surface-water. Consider a problem where the key issue with sustainable management is water-level declines from excessive pumping. The operational objective is to end up with an aquifer system where water storage does not change over the long-term while maintaining appropriate natural discharges to rivers and springs. Reductions in storage due to unsustainable production can only be reversed in two ways – increasing the quantity of inflows to the aquifer (e.g., recharge) or decreasing the outflows (e.g., pumping with wells). The yellow box in Text Box 2 lists four recharge schemes to increase inflows to aquifers (i.e., MAR) with links to the associated issues/problems, as indicated by the red arrows.

Clogging is a problem reducing the quantities of water infiltrated or injected into the subsurface, but can be managed with regular maintenance to maintain performance. In an Asian context, the other issues affecting MAR (Text Box 2) also provide formidable challenges. Finding water to recharge an aquifer can be difficult. Surface water can be scarce because excess water is often only available with summer monsoons. Treated municipal sewage, another important source of water, is often not available or of appropriate quality. For example, ~50% of Delhi's population has no sewers (Sengupta, 2015) with significant quantities of wastewater dumped into the nearby Yamuna River or left to seep into the ground. In addition, there tends to be declining interest in projects involving long transfers of water. Farmers in India (and China) prefer groundwater for irrigation as compared to government-supplied surface water (World Bank, 2010). Infrastructure, like reservoirs, pipelines or canals is needed to transfer water to where it is needed.

A variety of strategies exists to reduce groundwater withdrawals. Replacing groundwater (i, ii green, Textbox 2) in irrigation with imported surface water or treated wastewater is often challenging and would require re-imagining of water support systems. Decreasing agricultural production through acreage reductions, growing one crop per year instead of two, or changing to crops that use less water will lead to less groundwater utilization (iii to vi, Text Box 2). Yet, on the one hand, with governments firmly committed to food security and poor farmers needing to maintain their livelihoods, such initiatives are unattractive. On the other hand, these strategies require minimal technical expertise. Governments can pass a law, check the sustainability box, and plan to spend money to import some food. Finally, more efficient irrigation technologies might lead to reduced pumping, while also leading to reduced recharge (Garduno and Foster, 2010).

The feasibility of sustainable groundwater management is on display with projects of Orange County Water District (OCWD) in southern California. This case study illustrates that in arid areas with modest recharge and significant withdrawals, groundwater sustainability will require MAR. It also shows how wastewater recycling can provide a source of water when there are limited prospects for new surface-water. However, this source is expensive, in terms of physical infrastructure and advanced technologies for purification. OCWD distributes water to ~2.4 million people. Sustainable operation of the aquifer systems produces ~345 $Mm^3$ $yr^{-1}$ of groundwater, which is ~4.7 times the natural recharge of 74 $Mm^3$ $yr^{-1}$ (Hendron and Markus, 2014). MAR, using infiltration basins, makes up the deficit, with 185 $Mm^3$ $yr^{-1}$ coming from the Santa Ana River and 86 $Mm^3$ $yr^{-1}$ from purified urban wastewater (Hendron and Markus, 2014). Municipal wastewater is collected and treated conventionally, then purified with additional advanced treatment with reverse osmosis and more. Some of this purified water is used to maintain a hydraulic barrier in the subsurface to prevent seawater intrusion. This kind of system, providing evidence-based sustainability and high quality water, is expensive to build and operate. It is also critically dependent on monitoring (OCWD, 2015).

Such sophisticated water management systems are uncommon in Asia. Yet there are several extraordinary examples. The island state of Singapore is home for an innovate collection of management activities creating near self-sufficiency from water imports from Malaysia (Irvine et al., 2014). Drinking and industrial waters come from capturing and treating rainwater captured with urban catchments, the advanced purification of urban wastewater to a product called NEWater, and the addition of desalination plants (Irvine et al., 2014). MAR projects in Israel also provide other useful examples. The Dan Region

Reclamation Project (also known as Shafdan) uses treated wastewaters from Tel-Aviv and environs for MAR (Cikurel et al., 2012). The system yields 140 Mm³/yr of high quality water that is pumped 100 km south for irrigation. As of 2012, this was the largest project of its kind in Europe and the Middle East (Cickurel et al., 2012). Israel also depends on the reverse osmosis of seawater with periodic storage of excess water in the Israeli Coastal Aquifer (Ganot et al., 2018).

    The common characteristics of all three of these successful implementations include (i) extreme shortages of water to the point

of exhausting local surface water and groundwater supplies, (ii) technologically advanced and prosperous societies, with modern and reliable water/power infrastructures, and (iii) a manageable problem scope stemming from relatively small populations.

## 4 Groundwater Management: A Call to Action

There are compelling arguments to explain the slow development of a sustainability ethic in some Asian countries. In Pakistan, India and China, fragmented governance and the general inability to bring traditional socio-economic tools to bear on reducing groundwater demands impede progress to groundwater sustainability. The indictment for India, "centuries of mismanagement, political and institutional incompetence; indifference at central, state, and municipal levels, and steadily increasing population" (Biswas et al., 2017) applies as well to other countries. For India and Pakistan, where operational management is at the level

of states and provinces, there is no capacity to regulate. Also in both China and India, the tremendous numbers of groundwater users large and small confound regulation of groundwater. Thus, groundwater that has always been freely available for irrigators remains so.

    Groundwater-related problems are largely invisible (Biswas et al., 2017) and seemingly irrelevant to a greater agenda. It may also be that groundwater is so plentiful that it has never been a concern (Fogg, 2019).  For China, India, Pakistan and Iran,

there is an undeniable focus on food production to support growing populations and changing food preferences of increasingly affluent societies (Young et al., 2019). The continuing trend towards urbanization at all scales up to megacities is localizing water demands and exacerbating groundwater problems. The existence of necessary data as a prerequisite for problem understanding and management decisions remains a problem for all countries except perhaps China.

    There is, however, some hope that new technologies, may create sufficient visibility on the severity of the groundwater

problems to finally spur action (Fogg, 2019). For example, there are relatively inexpensive, wireless technologies available for monitoring water levels in real time. Expected improvements in satellite-remote sensing, particularly future GRACE missions, are also expected to enhance our understanding of aquifers worldwide (Fogg, 2019). However, field-based hydrogeological campaigns (e.g., MacDonald et al., 2016) and comprehensive water-quality monitoring will be necessary.

    With China as an obvious exception, groundwater-related research is not close to where it needs to be. Our review is evidently

cursory, focused mainly on available information in journal papers and government reports. Much of what is known about the groundwater in Asian countries (China again excepted) comes from international researchers but with in-country collaborators

and cooperators. Influential in this respect are long-term studies by The World Bank on specific strategies for management, large-scale interpretations of water storage changes with GRACE, and a few regional groundwater investigations. One area of critical need is research to address water quality issues in both India and Pakistan. Work by MacDonald et al. (2016) and Foster

et al. (2018) has identified the threat to sustainability related to problems of salinity and groundwater contamination. For example, with IGA aquifer system, this threat around issues of water quality is more serious than with over-pumping (MacDonald et al., 2016). Problems of arsenic pollution are widespread including Pakistan, while human activities that have led to groundwater salinization and urban/agricultural contamination (MacDonald et al., 2016). There are also hints of broad groundwater contamination in China (Biswas and Hartley, 2017), although information is scarce.

Adding water-quality issues to the mix of sustainability issues reveals even greater deficiencies and a need for water-quality monitoring as an essential step for sustainable management. Salinity problems are complicated because impacts can occur in so many ways. In Pakistan, saline water exists at depth in addition to salinized recharge caused by waterlogging. Moreover, this deep groundwater water can be remobilized by pumping (Foster et al., 2018). In China, shallow groundwater across the eastern half of the North China Plain is salinized (Foster and Garduno, 2004). This creates the possibility for eventual water

quality impairment in the underlying deep freshwater aquifer as over-pumping there continues.

We hope that countries in Asia begin to address sustainability problems more aggressively in critical areas, including the practical hydrogeologic investigations needed to support complex projects. There is some thinking that after many decades of aggressively exploiting aquifers, societies have begun to wake up to the need to manage and recharge aquifers (Fogg, 2019). Yet, there is a significant risk that progress will be slow in many Asian countries given the problems of capacity and socio-

economic constraints. A major transition will be required to move from water policies, viewed widely as muddling along from one crisis to the next without little substantive action to full-blown projects with demonstrated compliance (i.e., Singapore, California). Only time will tell, as to whether the successful water management schemes in places with relatively small and economically advantaged populations are practically scalable to many tens-of-millions of people in developing countries. In any case, logistical constraints mean that it will be decades before sustainable systems are up and running. Such a delay

increases the possibilities of predictable surprises – the problems (e.g., climate change) that are anticipated but ignored (Bazerman and Watkins, 2004).

There are basic technical approaches that have the potential contribute to sustainability. For example, several countries are already invested in recharge projects, India with their tradition MAR (Davis et al., 2018) and China with their "sponge city" concepts. Significant opportunities exist in identifying strengths and weaknesses in these methods, and in optimizing the

benefits for groundwater sustainability. To be most useful, studies should focus on best practices appropriate to the economic and technical capacities of the countries involved. While, these approaches are represent an important first step to groundwater sustainably, they are no panacea. For example, tradition approaches to water harvesting in India are not well suited for hard-rock areas, impact downstream users, and often lead to more pumping (World Bank, 2010). In addition, there is significant uncertainty as to whether these approaches will contribute meaningfully to sustainability, especially with uncontrolled

withdrawals. All of these technologies would benefit from analyses to identify strengths and weaknesses and to optimize the

benefits for groundwater sustainability. Studies sponsored by the World Bank (2010) suggest that there is hope for community-based management in the hard-rock areas of India and perhaps elsewhere.

There are risks that that Iran, India and Pakistan will run short of groundwater from over-pumping in some places and be adversely affected by global climate change, especially floods and droughts. The dimensions of these risks are not well defined.

Yet, in the case of climate change, a forward-looking study has examined the extent of in-country migration in order that countries "can plan and prepare" (Rigaud et al., 2018). The study involved modeling future migration in arid regions of the world due to climate change. The results for South Asia suggested that by 2050, there could be 35.7 million in-country climate migrants under a pessimistic future climate scenario. In the context of groundwater sustainability, we envision a need for similar scoping studies to examine the threats associated with running out of groundwater. No doubt, such analyses would be

difficult and uncertain, given absence of data. Yet, with potentially 100s-of-millions of people at risk, it would be prudent to better understand the scope and scale of future problems.

**Data availability.** The production of the digital elevation maps for Figure 2 used the following data sources:
 SRTM 90m DEM Version 4, Accessed and download from:  http://srtm.csi.cgiar.org/srtmdata/

GIS data for administrative area boundaries from: https://www.diva-gis.org/gdata
Country boundaries: https://www.naturalearthdata.com/downloads/50m-cultural-vectors/
(Last access for all 15 May 2019)

**Author contributions.** FS conceived the idea and wrote much of the manuscript. GL created the colored, three dimensional

elevation maps for China and India and prepared Figs. 1 and 2. Both GL and ZY contributed to the paper, especially insights and material with respect to China. All authors reviewed early manuscript drafts and the final draft.

**Competing interests.**

The authors declare that they have no conflict of interest.

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

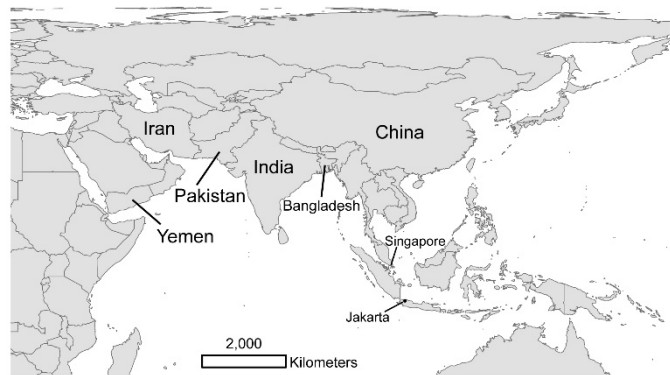

**Figure 1: Map showing the Asian countries and cities discussed in the paper.**

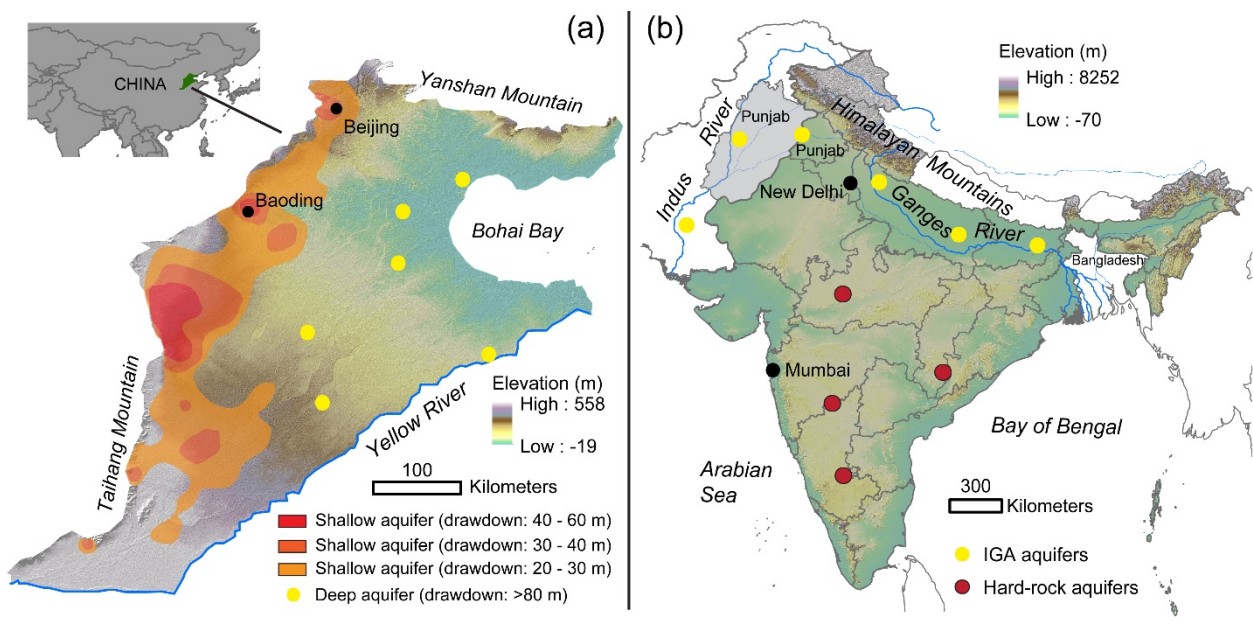

**Figure 2: Panel (a) is a shaded relief map of the core of the North China Plain. The red/orange colored areas have drawdowns >20 m in the shallow aquifer (Cao et al., 2012). The yellow dots indicate areas with local drawdowns in the deep aquifer >80 m (Zheng et al., 2012). Panel (b) shows India, Pakistan and Bangladesh. The red dots generally indicate the locations of hard-rock aquifers. The yellow dots point to the general location of IGA system along the plains of the Indus and Ganges Rivers.**


### Delhi's great water fall: Capital fears riots and water shortages as groundwater level hits dangerous low

- S and SW Delhi water table declined 10-20 m last 10 yrs
- At 20-50 m water brackish or saline
- Contamination by $NO_3$, F  (Sharma, 2013)

### Delhi groundwater, a deadly cocktail

- 42 of 124 samples salinity 2000 to 16,700 uS/cm
- 6 of 122 $NO_3$ 800-1500 mg/L
- 29 of 122 $NO_3$ 100 -800 mg/L  (Seth, 2015)

### Groundwater Plummets in Delhi, City of 29 Million

- Could reach "zero groundwater levels" by 2020
- M $m^3$/yr groundwater pumped vs. natural recharge 310 M $m^3$/yr
- groundwater levels critical in 90% of city and demand is growing  (Ritter, 2019)

**Text Box 1: Headlines and articles reacting to Delhi's worsening groundwater problems.**

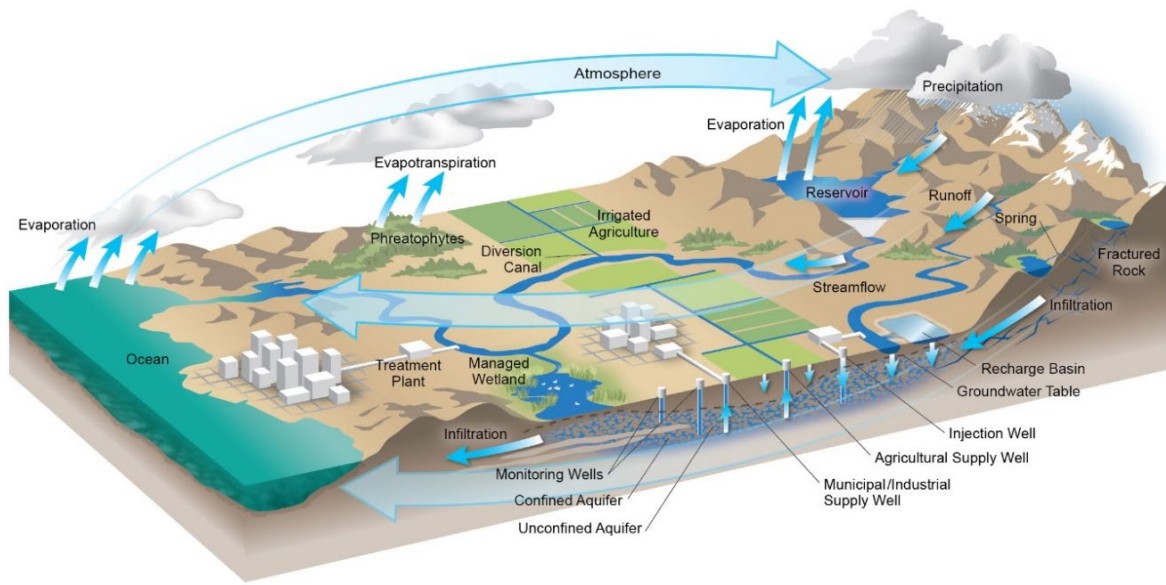

540

**Figure 3: Conceptual model of a hypothetical coastal hydrologic system featuring a major river, cities, agricultural irrigation, an alluvial aquifer system being recharged using various MAR systems, and more. Complexities arise from (i) the number of different processes operating within the basin and their associated parameters, (ii) a need to quantify the diverse array of water exchanges within the hydrologic cycle, (iii) water uses that need to be metered, (iv) potential groundwater contamination from irrigation return flows, and (v) constraints dictated by sustainable groundwater management. (With permission, California Department of Water Resources, 2016).**

545

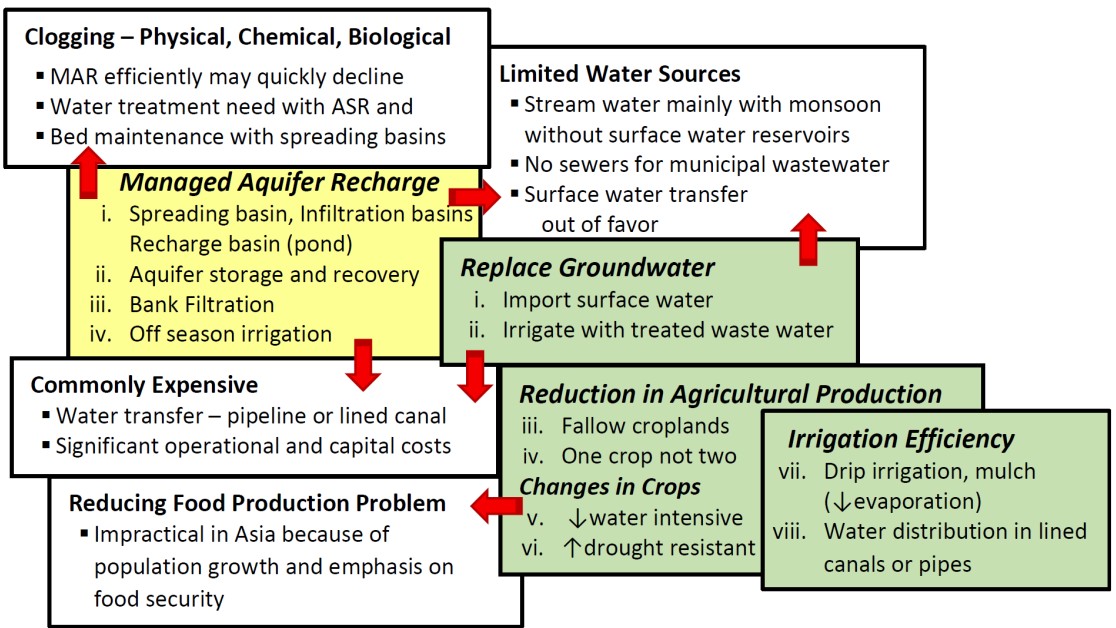

**Clogging – Physical, Chemical, Biological**
- MAR efficiently may quickly decline
- Water treatment need with ASR and
- Bed maintenance with spreading basins

**Limited Water Sources**
- Stream water mainly with monsoon without surface water reservoirs
- No sewers for municipal wastewater
- Surface water transfer out of favor

*Managed Aquifer Recharge*
  i. Spreading basin, Infiltration basins Recharge basin (pond)
  ii. Aquifer storage and recovery
  iii. Bank Filtration
  iv. Off season irrigation

*Replace Groundwater*
  i. Import surface water
  ii. Irrigate with treated waste water

**Commonly Expensive**
- Water transfer – pipeline or lined canal
- Significant operational and capital costs

*Reduction in Agricultural Production*
  iii. Fallow croplands
  iv. One crop not two

*Changes in Crops*
  v. ↓water intensive
  vi. ↑drought resistant

*Irrigation Efficiency*
  vii. Drip irrigation, mulch (↓evaporation)
  viii. Water distribution in lined canals or pipes

**Reducing Food Production Problem**
- Impractical in Asia because of population growth and emphasis on food security

**Text Box: 2. The yellow and green boxes list some of the strategies for increasing aquifer storage of groundwater by increasing inflows through managed recharge and/or decreasing the quantity of water pumped, respectively. The red arrows indicate associated**
 **issues.**