# Peer review of "HESS Opinions: The Myth of Groundwater Sustainability in Asia"

_Hydrology and Earth System Sciences, 2019_

## Referee Comment (RC1) · Anonymous Referee #1 · 20 Aug 2019

The opinion paper by Frank Schwartz and coauthors discusses the lingering groundwater crisis in several Asian countries, some reasons how it could come so far, theoretically feasible technical solutions, and vague research directives.

It is clear, that groundwater exploitation is not sustainable in many countries with (semi)-arid climate, including actually large parts of the United States. However, besides climate and land use there are also societal boundary conditions, and these differ tremendously between the countries discussed in the manuscript. The People's Republic of China definitely does lack democratic participation, but it has a long standing tradition of a functional administration, and the economic growth of the last decades has led to the economic foundation for expensive technical solutions, if applicable. We see this in water treatment (both for freshwater and waste water) where tremendous

progress has been made in recent years. Of all countries discussed in the manuscript, China is the one where the educational and administrative conditions are the best to implement water-management strategies comparable to those of Southern California - if the Communist Party decides sustainable groundwater management to be an important issue.

In contrast, other countries lack the concept of groundwater rights. If traditionally the owner of a piece of property is allowed to extract all resources thereof, including groundwater, implementing rules of sustainable groundwater management is doomed to fail. There must be an accepted legal framework stating that you don't own the water of the land that you own, that drilling and operating a new well requires a permit, that the permit can only be issued based on a management plan of the entire resource, that abiding by the rules must be monitored, and that a breach of regulations must be punished. If this basic societal understanding does not exist, sustainability cannot be enforced.

I don't think that the authors should put Yemen into the mix of countries to consider. Yemen has been in a Civil War for years, and one cannot expect that anything functions. Almost the same would hold for Afghanistan where the German Geological Survey had spent millions on developing groundwater management rules, including hydrogeological mapping and implementing groundwater monitoring. All of that disappeared when the security of western advisors was no more guaranteed. In such dysfunctional countries, sustainable groundwater management cannot be of high priority. Whereas it could in India.

The authors present Orange County and Singapore as highly developed regions in which technical solutions for sustainable groundwater management have more or less successfully been implemented, monitored, and maintained. They could add Israel where advanced irrigation techniques and managed aquifer recharge has been developed on a world leading level. Like in Singapore, if even not much more so, Israel is in need of self-sufficiency, has a functional administration, and is home of some of the best engineers worldwide. Hence, when it comes to discussing why sustainable

groundwater management appears achievable in Israel but not so much in some of its neighboring countries with similar climate and geology, the societal and governmental boundary conditions must be analyzed to a depth at which geologists and engineers feel uncomfortable. Being a hard-core scientist myself, I lack an in-depth discussion of societal differences among the different countries that can explain differences and give predictions on the chances of implementing sustainable groundwater management practices. Iran, India, China, and Pakistan are quite different countries.

The authors rightfully point to water-quality issues related to groundwater management in arid climates and/or regions of intensive agriculture. However, you don't need to go to Asia to realize that salt accumulation in over-exploited aquifers is an issue largely unrecognized by many groundwater managers. In large parts of the western United States, a continuous increase in salinity has been observed in conjunction with declining groundwater levels. At the end of the day, balancing the volume of water is insufficient to obtain sustainability in systems undergoing strong evapotranspiration. We may come to the conclusion that managing the dissolved solids will require more aggressive treatments, such as membrane-based deionization before artificial groundwater enrichment. Luckily, the electricity needed for that can be gained by photovoltaic power in the arid regions that require such treatments the most. Likewise, arsenic (or fluorine) can be removed by technical treatment, but the premise of centralized water treatment is a centralized water supply. In as much, technical solutions for the supply of cities, where centralized treatment options are achievable, must differ from technical solutions for drinking water supply and irrigation agriculture in rural regions. And neither will work without a functional and responsible administration.

With respect to research directives, I highly recommend prioritization. Western researchers are interested in exciting science, but that is not always the gateway to practical solutions. Understanding the release and fate of arsenic in deltaic aquifers in south-east Asia is an example of a scientifically challenging question. Alas, among the hundreds to thousands of publications on mechanistic questions related to arsenic in south-east Asia, only a few have been useful to help the people affected. There have

been examples in which "cool" science actually contributed to developing sustainable groundwater management strategies, but most of the science is done by the flock of academic sheep following a research bellwether. Most likely, raising the level of education in water-related sciences is the best that university scientists can do to contribute; we need to train people with a solid understanding of hydrogeology and environmental engineering, who hopefully reach positions where they can make decisions. But how a society has to change that responsible decision making by administrative authorities is implemented and accepted, I have no clue.

A few minor comments.

1. line 33: Replace "by right" with "basically". Non-native speakers think you refer to a legal term.

2. lines 43-44: Are there only one continuous shallow and one continuous deep aquifer in the entire North China Plain? Otherwise use the plural.

3. line 58: Do the percentages refer to India or are the worldwide numbers? The same question refers to the "two prototypical settings for groundwater".

4. line 63: "recover to the levels pf previous years" or "recover from the withdrawals of previous years."

5. line 77: The term "regionalized" appears odd here. This is a term used in geostatistics for interpolation of point data, but it seems you mean "restricted to certain regions".

6. line 81: While the root cause of arsenic in the IGA system is in the Himalayan sediments, the mechanism are more complicated. I suggest dropping this explanation in order to avoid oversimplification.

7. line 92: Nitrate is sometimes measured as concentration nitrate, and sometimes as concentration nitrate-N. Be specific!

8. line 113: I don't think that the assessments are frustrated. Frustrated are the people performing the assessment, which renders the assessment frustrating.

9. line 139: drop "landward", seawater intrusion is always landward.

10. lines 150-156: I don't think that it is admissible to compare the situation in India (a functional government) to that in Yemen (Civil war)

11. line 174: Here is one of the points. You cannot assume that appropriate laws and regulations exist in all countries that need them. That is exactly the point.

12. lines 309: The authors may check on the work of Wolfgang Kinzelbach (who is not me, by the way), who has worked on stretching groundwater exploitation in northern Africa, a definitely non-sustainable resource.

---

## Referee Comment (RC2) · Anonymous Referee #2 · 3 Sep 2019

This is an interesting opinion paper on a well-known and significant topic. I enjoyed reading it, especially the review on the case studies and the main problems hampering the effective and sustainable management of groundwater resources. To my best knowledge, the "myth" of groundwater sustainability, and groundwater management in general, belong to many countries, even "advanced" ones, not only Asian.

The paper is made of two parts: illustration of selected examples and some proposals for a "pragmatic research agenda". The first part is quite good and convincing, although the main conclusions are unfortunately rather obvious and well known nowadays. The collection of cases is not a comprehensive review of groundwater management case in Asia, and it is not meant to be that, but it delivers the message; still, the socio-political conditions are much different among sites such that a comparison is not possible.

Perhaps the main focus of the hurdles is on the technical issues, less on the socio-political constraints that in many cases lead the process. My main reservation is that the exposition looks confusing at times. For instance, the examples continue in Section 3 (by the way, the case of Yemen seems to me quite divorced from the rest standing the particular situation of the area) and one cannot truly see a discontinuity between sections 2 and 3. The lengthy text on the OCWD seems quite out of place and not in line with the rest, which focuses on Asian countries (and do we need Eq.1?). A few sentences would have delivered the same concept. Similar for the Singapore case.

The second part, i.e. the delineation of the proposed ideas based on the current management practice in Asia, is much shorter than the first one and not much clear in my view. It definitely needs more elaboration. The Section promises "Groundwater Research Directions" but I can't really find clear and sufficiently elaborated indications.

The first item deals with water quality; adding water quality to the management practices seems rather obvious, and it is implicitly done in several cases, but perhaps I have misunderstood the point (and the short text does not help).

I agree in principle with the approach of considering the sustainable groundwater management as something that will never materialize, and the derived idea of the worst-case scenario. This is something interesting and useful, and sometimes I have seen a similar approach adopted in practical management schemes. However, I see two problems with this approach. First, the analysis of the worst case scenario may anyway need significant resources for data acquisition and the understanding of the groundwater-surface water interactions, and then the several technical problems illustrated in the paper come back again. Second, the message that may easily come out from this suggestion is the following: forget about management, too difficult and expensive, just let things go and prepare for the worst. That would mean the death of the concept of sustainable management and the triumph of Business As Usual, with likely disastrous consequences on areas characterized by poor or absent management.

[Figure]

Instead, I think that a less pessimistic alternative would be to provide a management procedure made by subsequent steps of increasing complexity, starting from basic and simple analyses that may guide the management and political decision; in other words, not give up the concept of management. In this perspective, one would rather speak of "feasible management", i.e. based on analyses that can be realistically carried out under the several constraints, starting from the simple concept of safe yield that is relatively easy to estimate in most cases. The governments and stakeholders may start making decision (import food? Invest more on different sources of water? etc.) from those basic and anyway fundamental pieces of information. Role of the scientists and engineers is to try to provide simple rules to stakeholders and managers, while complex management techniques may be affordable only by California or a few other developed regions. To this matter, the list of technical requirements brought by the paper is certainly discouraging. Thus, while the worst case scenario is something worth performing (but how about its uncertainty? Are the future stressors certain?), giving up completely the idea of management might not be so good. Again, I might have misunderstood the concept, and this part of the paper (Section 4) needs further clarification and elaboration.

---

## Referee Comment (RC3) · Graham Fogg (Referee) · 27 Sep 2019

This Opinion paper is a well-written, sobering description of the ongoing crisis of groundwater mis-management in Asia and prospects for changing course. Despite its negative bottomline message that the crisis likely cannot be averted, I enjoyed reading the paper and believe the readership will find it interesting and thought provoking.

All of my edits and comments are marked directly in the PDF that is uploaded with this review.

My main comment is that the message – that it's highly unlikely for groundwater in Asia to ever be managed sustainably – is too negative. Granted, this is an opinion piece, and the authors are entitled to their opinion, but I think they might be missing an opportunity

to provide more impetus for positive change. I worry that the negative message may do more to stifle groundwater management than to produce beneficial change, and all under the assumption that such change is impossible. For added perspective, consider the following:

- Any of the needed groundwater information infrastructure would be cheap relative to the spending these countries are currently doing for construction and maintenance of surface water infrastructure (dams and conveyance). So if they realize they must have something, they can likely find the means to achieve it. One less dam project could free up enough funds for a national groundwater monitoring network. Thailand's Department of Groundwater (yes, there is such a thing) has been doing this nationally since the 1950-60s and hence has been more proactively managing groundwater.

- The world may be entering a period of change with respect to groundwater management, although it may require considerable coaxing and crises to get there. Since widespread deployment of industrial scale groundwater pumping technologies some 70 yrs ago, very little effort has been devoted to recharging and managing groundwater. In essence, civilization has not yet begun to try to manage groundwater very much, mainly because it has not had to, mainly because of the vastness of most groundwater basin resources. But now that may be starting to change. See the discussion piece: https://trend.pewtrusts.org/en/archive/spring-2019/groundwater-the-resource-we-cant-see-but-increasingly-rely-upon . I agree – it is questionable whether such change can happen soon enough in Asia, and people should also start preparing for the worst.

-One could argue that a big part of the problem is the lack of transparency of groundwater systems, making the state of gw resources easier to ignore. There are technologies coming along that could change this significantly – e.g., low-cost wireless, real-time groundwater level monitoring networks connected to open-source web platforms to track fluctuations in groundwater levels (these may require cellular networks, which are already more extensive in parts of rural Asian than parts of rural America); and future

versions of GRACE that work on spatial scales of 50 km rather than 400 km (National Academies of Sciences, Engineering, and Medicine 2018. Thriving on Our Changing Planet: A Decadal Strategy for Earth Observation from Space. Washington, DC: The National Academies Press. https://doi.org/10.17226/24938.) If a new generation of GRACE comes along that actually works on relevant water resources management scales (∼50km), that could be revolutionary for groundwater management. The satellite deployment agencies across the globe need to be reminded of this constantly to help make it happen.

I still recommend the paper be published with minor changes, but just suggest you reconsider the tone and bottomline message in light of the above.

Best Wishes, Graham Fogg

Please also note the supplement to this comment:
https://www.hydrol-earth-syst-sci-discuss.net/hess-2019-399/hess-2019-399-RC3-supplement.pdf

**Supplement:**

**HESS Opinions: The Myth of Groundwater Sustainability in Asia**

Franklin W. Schwartz[1], Ganming Liu[2] and Zhongbo Yu[3]

[revised manuscript text omitted]

Number: 1 Author: Subject: Cross-Out Date: 9/26/19, 11:06:48 AM

Number: 2 Author: Subject: Inserted Text Date: 9/26/19, 11:09:49 AM
Had decision-makers acted on the basis of scientific knowledge

Number: 3 Author: Subject: Inserted Text Date: 9/26/19, 11:10:38 AM
would

Number: 4 Author: Subject: Cross-Out Date: 9/26/19, 11:07:20 AM

Number: 5 Author: Subject: Sticky Note Date: 9/26/19, 11:12:11 AM
Perhaps add a sentence pointing to possible reasons why the mere existence of knowledge about the problem did not lead to actions.

Number: 6 Author: Subject: Inserted Text Date: 9/26/19, 11:16:02 AM
to

Number: 7 Author: Subject: Inserted Text Date: 9/26/19, 11:16:37 AM
to

Number: 8 Author: Subject: Inserted Text Date: 9/26/19, 11:15:49 AM
from

[revised manuscript text omitted]

Number: 1 Author:     Subject: Inserted Text          Date: 9/26/19, 11:45:14 AM

use

Number: 2 Author:     Subject: Inserted Text          Date: 9/26/19, 11:45:35 AM

annual?

Number: 3 Author:     Subject: Highlight     Date: 9/26/19, 11:50:25 AM

Sentence too negative. The MAR is an incredibly positive thing, but it's just that one also must manage the pumping.

Number: 4 Author:     Subject: Inserted Text          Date: 9/26/19, 11:49:13 AM

without actions that also manage groundwater demand and the actual amounts pumped

Number: 5 Author:     Subject: Inserted Text          Date: 9/26/19, 11:50:59 AM

as

Number: 6 Author:     Subject: Inserted Text          Date: 9/26/19, 11:51:33 AM

and

[revised manuscript text omitted]

Number: 1 Author: Subject: Inserted Text Date: 9/26/19, 12:10:47 PM

use,

Number: 2 Author: Subject: Highlight Date: 9/26/19, 12:19:02 PM

I think you need to be more nuanced about cost. Compared to spending almost $0 on groundwater information infrastructure, it will certainly cost and be difficult in countries that are not wealthy. However, compared to building, maintaining and operating surface water infrastructure (e.g., dams and canals), the cost of the needed groundwater information infrastructure is small. These countries are engaged in some major dam-building projects, while far the biggest 'reservoirs' and 'spaces' for storing water lie under their feet.

Number: 3 Author: Subject: Highlight Date: 9/26/19, 12:22:14 PM

Yes, but scrapping fines off spreading basins is not very expensive -- much cheaper than maintaining a road.

Number: 4 Author: Subject: Cross-Out Date: 9/26/19, 12:23:32 PM

Number: 5 Author: Subject: Highlight Date: 9/26/19, 12:27:09 PM

This is arguable. Once some of the information infrastructure that you mention above is in place, it will become clearer to people the consequences of not managing groundwater, including consequences of not conveying water to aquifers for subsurface storage etc. So 'not practicable' seems to strong. 'Challenging and requiring major re-imagining of the water support systems' might be more appropriate.

will lead to less groundwater utilization (iii to vi, Text Box 2). Yet, on the one hand, with governments firmly committed to food security and poor farmers needing to maintain their livelihoods, such initiatives are unattractive. On the other hand, these strategies require minimal technical expertise. Governments can pass a law, check the sustainability box, and plan to
235    spend money to import some food. Finally, more efficient irrigation technologies might lead reduced pumping (Garduno and Foster, 2010) 2

There are successful models for managing water resources sustainably. They are worth discussing here to illustrate (i) requirements for data and advanced technologies, (ii) the long-term commitment to complex and costly projects, and (iii) efforts necessary to turn urban wastewater into a valuable water source. The Orange County Water District (OCWD) in
240    southern California near Los Angeles is a leader in sustainable groundwater management. OCWD serves a 900 km$^2$ area, distributing water to ~2.4 million people. Two thirds of that water comes from groundwater produced from hundreds of deep, high capacity wells. Sustainable operation of the aquifer systems provides ~345 Mm$^3$ yr$^{-1}$ of high-quality groundwater, while maintaining aquifer storage within a specified operating range (Hendron and Markus, 2014). With natural recharge of only 74 Mm$^3$ yr$^{-1}$, sustainable management requires additions to storage through MAR (Hendron and Markus, 2014). Water
245    is diverted from the Santa Ana River and infiltrated through recharge basins (Figure 4). River flow is mostly treated sewage and occasional winter stormflows.

Another recharge source is purified water produced by the Groundwater Replenishment System (GWRS) facility (Figure 4). Municipal wastewater, collected and treated at the Orange County Sanitation District (OCSD) treatment facility, is transferred to the GWRS facility for advanced treatment (Hendron and Markus, 2014). Processing that includes reverse
250    osmosis and other treatments produces near-distilled water. Approximately 65% of this water moves through pipes to recharge basins at Anaheim (follow the red line, Figure 4). The remainder is injected through a line of wells completed at various depths, creating a seawater intrusion barrier.

A simple water-balance (Eqn. 1), based on Hendron and Markus (2014), shows pumped groundwater (PGW) to be balanced by natural recharge (NR), MAR using both infiltrated river water (IR) and advanced treated wastewater (ATW):

255

$$\text{ Mm}^3 \text{ yr}^{-1}{}_{NR} + 185 \text{ Mm}^3 \text{ yr}^{-1}{}_{IR} + 86 \text{ Mm}^3 \text{ yr}^{-1}{}_{ATW} - 345 \text{ Mm}^3 \text{ yr}^{-1}{}_{PGW} = 0 \qquad (1)$$

The success of OCWD's hydrogeological operations is critically dependent on monitoring. They collect production data monthly for the high capacity production wells and less frequently for smaller wells (OCWD, 2015). Water level and water
260    quality data coming from hundreds of wells provides evidenced-based compliance with sustainability goals. The quality of water from the GWRS facility is monitored, as is the Santa Ana River and tributaries (OCWD, 2015). Performance of the seawater intrusion barrier is also monitored along with subsidence across the basin.

Such sophisticated water management systems are uncommon in Asia. Yet, the island state of Singapore is home for an innovate collection of management activities creating near self-sufficiency from water imports from Malaysia (Irvine et al.,
265    2014). Drinking and industrial waters come from capturing and treating rainwater captured with urban catchments, the

Number: 1 Author:     Subject: Inserted Text          Date: 9/26/19, 12:27:44 PM
 to

Number: 2 Author:     Subject: Inserted Text          Date: 9/26/19, 12:28:14 PM
, while also leading to reduced recharge.

advanced purification of urban wastewater to a product called NEWater, and the addition of desalination plants (Irvine et al., 2014).

**[1] Planning for the Worst and Hoping for the Best: Groundwater Research Directions**

270 There are compelling arguments why it is unrealistic to expect groundwater to be managed sustainability in developing Asian countries. The indictment for India, "centuries of mismanagement, political and institutional incompetence; indifference at central, state, and municipal levels, and steadily increasing population" (Biswas et al., 2017) applies as well to other countries. Groundwater-related problems are largely invisible (Biswas et al., 2017) and seemingly irrelevant to a greater agenda. For China, India, and Iran, there is an undeniable focus on food production to support growing populations

275 and changing food preferences of increasingly affluent societies (Young et al., 2019). The continuing trend towards[2] urbanization at all scales up to megacities is localizing water demands and exacerbating groundwater problems.

Despite progress with satellite-remote sensing, particularly GRACE (Feng et al., 2018; Long et al., 2017; Rodell et al., 2009; Rodell et al., 2018), actions around evidence-based groundwater sustainability is at an early stage. In the case of the IGA aquifer system, the greatest present threat to long-term sustainability is not from over-pumping but from human activities

280 that have led to groundwater salinization and urban/agricultural contamination (MacDonald et al., 2016). This experience in India and Pakistan and possibly China reveals how pervasive contamination can lead to the same unsustainable outcome as over-pumping.

Adding water-quality issues to the sustainability mix reveals even greater deficiencies in data and needs for research in modeling and arid-zone geochemistry. For example, salinity problems are complicated because impacts can occur in so

285 many ways. In Pakistan, saline water exists at depth in addition to salinized recharge caused by waterlogging. Moreover, this deep groundwater water can be remobilized by pumping (Foster et al., 2018). In China, shallow groundwater across the eastern half of the North China Plain is salinized (Foster and Garduno, 2004). This creates the possibility for eventual water quality impairment in the underlying deep freshwater aquifer as over-pumping there continues. Research is required to explore mechanisms, pathways and time scales of contaminant-related impacts on sustainability of aquifers. Another target

290 of opportunity is the difficult field characterizations of the geochemistry of saline groundwaters in arid-zone settings.

A pragmatic research agenda must also account for the risk that sustainable groundwater management will never happen. The necessary transition from a water policy of muddling along, stumbling from one crisis to the next without substantive actions to quantifiably sustainable systems, like those in Orange County or Singapore will be enormous. Further, it is doubtful whether the successful strategies in those two places with relatively small and economically advantaged populations

295 are practically scalable to many millions of people in developing countries. In any case, logistical constraints mean that it will be decades before sustainable systems are up and running. Such a delay increases the possibilities of predictable surprises – the problems (e.g., climate change) that are anticipated but ignored (Bazerman and Watkins, 2004).

Number: 1 Author:    Subject: Highlight    Date: 9/26/19, 2:06:33 PM

Although there is a fair amount of wisdom in this closing section, I worry that its level of negativity may do more to stifle action than to spur action. One could argue that a big part of the problem is the lack of transparency of groundwater systems, making the system state of gw resources easier to ignore. There are low-cost technologies coming along that could change this significantly -- e.g., wireless groundwater level monitoring networks connected to open-source web platforms to track fluctuations in groundwater levels; and future versions of GRACE that work on spatial scales of 50 km rather than 400 km (National Academies of Sciences, Engineering, and Medicine 2018. *Thriving on Our Changing Planet: A Decadal Strategy for Earth Observation from Space.* Washington, DC: The National Academies Press. https://doi.org/10.17226/24938.)  Furthermore, since widespread deployment of industrial scale groundwater pumping technologies some 70 yrs ago, very little effort has been devoted to recharging and managing groundwater. In essence, civilization has not yet begun to try to manage groundwater, mainly because it has not had to, mainly because of the vastness of most groundwater basin resources. But now that is changing. See 
[revised manuscript text omitted]

---

## Author Comment (AC1) · 20 Oct 2019

We are appreciative of the constructive comments on the paper from all three reviewers. Following here is our detailed response to comments of Reviewer #1.

We have considered Reviewer #1's comments carefully and made significant changes to the paper.

Reviewer #1:

The opinion paper by Frank Schwartz and coauthors discusses the lingering groundwater crisis in several Asian countries, some reasons how it could come so far, theoretically feasible technical solutions, and vague research directives. It is clear, that groundwater exploitation is not sustainable in many countries with (semi)-arid climate,

including actually large parts of the United States.

1. However, besides climate and land use there are also societal boundary conditions, and these differ tremendously between the countries discussed in the manuscript.

(response) The original paper draft was focused on mainly technical issues that we considered as hurdles that needed to be overcome for quantitative and verifiable management of large aquifers. Our view was that these issues for many countries in Asia constituted barriers that by themselves would preclude serious efforts towards sustainability. In this respect, the availability of data represents a critical information gap for many countries because you cannot manage something you don't understand. We also used experience from Orange County, California and California more generally to illustrate the true challenges of sustainable management of groundwater from a technical perspective.

However, Reviewer #1 quite rightly pointed out that sustainable management also requires a proper legal and socio-economic framework for action. Our paper hinted at the necessity for laws as basis for enforcing limits on withdrawals and synchronization of macroeconomic policies, but the coverage was minimal. Following the reviewer's suggestion have we expanded the paper significantly to explain frameworks for action and the various components that contribute to sustainability, and to provide a context for key countries that we considered.

We address points 1 and 3 together by adding a long section [lines 38 to 66] in the revised introduction that describes robust frameworks shown to work in areas of legislation, policy, regulatory/ macroeconomic tools. In section "3. What are the Hurdles to Groundwater Sustainability?" we have rewritten and generalized the 2nd "hurdle" concerning data to describe the status of India, Pakistan, and China [lines 194 to 224] with respect to the socio-economic framework discussed in the introduction. The treatment is economical (adding $\frac{1}{2}$ page new) and a significant rewrite of associated material.

2. The People's Republic of China definitely does lack democratic participation, but it

has a long standing tradition of a functional administration, and the economic growth of the last decades has led to the economic foundation for expensive technical solutions, if applicable. We see this in water treatment (both for freshwater and waste water) where tremendous progress has been made in recent years. Of all countries discussed in the manuscript, China is the one where the educational and administrative conditions are the best to implement water-management strategies comparable to those of Southern California - if the Communist Party decides sustainable groundwater management to be an important issue. In contrast, other countries lack the concept of groundwater rights.

(response) We have added a section explaining the present status of groundwater management in China as well as Pakistan and India. The China piece is part of the longer section described in 1 above (3rd paragraph). Assessments by various authors indicate much slower progress in groundwater management than with surface water. We added the sentiment expressed by Reviewer #1 that they would have the financial and technical capacities if the government wished to make progress.

3. If traditionally the owner of a piece of property is allowed to extract all resources thereof, including groundwater, implementing rules of sustainable groundwater management is doomed to fail. There must be an accepted legal framework stating that you don't own the water of the land that you own, that drilling and operating a new well requires a permit, that the permit can only be issued based on a management plan of the entire resource, that abiding by the rules must be monitored, and that a breach of regulations must be punished. If this basic societal understanding does not exist, sustainability cannot be enforced.

(response) We agree with this point. As mentioned, this is a specific example of the problems discussed in 1 and has been addressed in the revision. The specific case of China is now described in much greater detail.

4. I don't think that the authors should put Yemen into the mix of countries to consider. Yemen has been in a Civil War for years, and one cannot expect that anything functions. Almost the same would hold for Afghanistan where the German Geological Survey had spent millions on developing groundwater management rules, including hydrogeological mapping and implementing groundwater monitoring. All of that disappeared when the security of western advisors was no more guaranteed. In such dysfunctional countries, sustainable groundwater management cannot be of high priority. Whereas it could in India.

(response) Rev#1 (and Rev#2) both recommended that we remove this piece and we have done so.

5. The authors present Orange County and Singapore as highly developed regions in which technical solutions for sustainable groundwater management have more or less successfully been implemented, monitored, and maintained. They could add Israel where advanced irrigation techniques and managed aquifer recharge has been developed on a world leading level. Like in Singapore, if even not much more so, Israel is in need of self-sufficiency, has a functional administration, and is home of some of the best engineers worldwide. Hence, when it comes to discussing why sustainable groundwater management appears achievable in Israel but not so much in some of its neighboring countries with similar climate and geology, the societal and governmental boundary conditions must be analyzed to a depth at which geologists and engineers feel uncomfortable. Being a hard-core scientist myself, I lack an in-depth discussion of societal differences among the different countries that can explain differences and give predictions on the chances of implementing sustainable groundwater management practices. Iran, India, China, and Pakistan are quite different countries.

(response) We agree with the reviewer's point here Israel is certainly worth noting as a country with success in managed aquifer recharge. We have added sentences to the discussion at this point in the paper discussing accomplishments in Israel. In our revision, we have pointed out our view as to why Orange County, Singapore, and Israel have been successful.

(revised wording in next draft). "Such sophisticated water management systems are uncommon in Asia. Yet there are several extraordinary examples. The island state of Singapore is home for an innovate collection of management activities creating near self-sufficiency from water imports from Malaysia (Irvine et al., 2014). Drinking and industrial waters come from capturing and treating rainwater captured with urban catchments, the advanced purification of urban wastewater to a product called NEWater, and the addition of desalination plants (Irvine et al., 2014). MAR projects in Israel also provide useful examples. The Dan Region Reclamation Project (also known as Shafdan) uses treated wastewaters from Tel-Aviv and environs for MAR (Cikurel et al., 2012). The system yields 140 Mm3/yr of high quality water that is pumped 100 km south for irrigation. As of 2012, this was the largest project of its kind in Europe and the Middle East (Cikurel et al., 2012). Israel also depends on the reverse osmosis of seawater with periodic storage of excess water in the Israeli Coastal Aquifer (Ganot et al., 2018).

The common characteristics of all three of these successful implementations include (i) extreme shortages of water to the point of exhausting local surface water and groundwater supplies, (ii) technologically advanced and prosperous societies, with modern and reliable infrastructures, and (iii) a manageable problem scope stemming from relatively small populations.."

We agree with the reviewers comments in the last few sentences of 5. As mentioned in 1 and 3 we have provided a much improved analysis of the legislative and operational "boundary conditions" to provide a better sense as to which countries are likely to succeed.

6. The authors rightfully point to water-quality issues related to groundwater management in arid climates and/or regions of intensive agriculture. However, you don't need to go to Asia to realize that salt accumulation in over-exploited aquifers is an issue largely unrecognized by many groundwater managers. In large parts of the western United States, a continuous increase in salinity has been observed in conjunction with declining groundwater levels. At the end of the day, balancing the volume of water

is insufficient to obtain sustainability in systems undergoing strong evapotranspiration. We may come to the conclusion that managing the dissolved solids will require more aggressive treatments, such as membrane-based deionization before artificial ground-water enrichment. Luckily, the electricity needed for that can be gained by photovoltaic power in the arid regions that require such treatments the most. Likewise, arsenic (or fluorine) can be removed by technical treatment, but the premise of centralized water treatment is a centralized water supply. In as much, technical solutions for the supply of cities, where centralized treatment options are achievable, must differ from technical solutions for drinking water supply and irrigation agriculture in rural regions. And neither will work without a functional and responsible administration. The paper already makes clear there is more to sustainability than taking care of water balances. Indeed this is evident as we mention in both India and Pakistan.

(response) We thank the reviewer for mentioning possibilities with dealing more aggressively with the water quality problems. Although the points raised here concerning remediation membrane-based deionization, arsenic removal, are interesting they might come much further in the future once sustainability problems are recognized and have begun to be dealt with. These are topics that we feel are beyond the scope of the present paper and significantly outside of our areas expertise. What we have tried to emphasize is that character, distribution and concentration of contaminants remains an informational black hole for all these countries including China. So, we made no changes in response to this point.

7. With respect to research directives, I highly recommend prioritization. Western researchers are interested in exciting science, but that is not always the gateway to practical solutions. Understanding the release and fate of arsenic in deltaic aquifers in south-east Asia is an example of a scientifically challenging question. Alas, among the hundreds to thousands of publications on mechanistic questions related to arsenic in south-east Asia, only a few have been useful to help the people affected. There have been examples in which "cool" science actually contributed to developing sustainable

groundwater management strategies, but most of the science is done by the flock of academic sheep following a research bellwether. Most likely, raising the level of education in water-related sciences is the best that university scientists can do to contribute; we need to train people with a solid understanding of hydrogeology and environmental engineering, who hopefully reach positions where they can make decisions. But how a society has to change that responsible decision making by administrative authorities is implemented and accepted, I have no clue.

(response) We certainly agree with these comments. Obviously, the scope and scale of existing and future problems are too serious to be poking around answering basic-science questions. We have both reworded and added sentences in the conclusion to reflect this view from Rev#1 as follows.

(revised wording next draft) "It is worthwhile to consider international research to support those sustainability initiatives underway and likely to continue. For example, countries appear to invest in recharge projects, India with their tradition MAR (Davis et al., 2018) and China with their "sponge city" concepts. There are significant opportunities in adapting modern analytical approaches to these various strategies to identify strengths and weaknesses and to optimize the benefits for groundwater sustainability. To be most useful, such research should focus on best practices appropriate to the economic and technical capacities of the countries involved."

A few minor comments.

1. line 33: Replace "by right" with "basically". Non-native speakers think you refer to a legal term. DONE

2. lines 43-44: Are there only one continuous shallow and one continuous deep aquifer in the entire North China Plain? Otherwise use the plural. OK as is

3. line 58: Do the percentages refer to India or are the worldwide numbers? The same question refers to the "two prototypical settings for groundwater". Word "India" added

twice in clarification

4. line 63: "recover to the levels pf previous years" or "recover from the withdrawals of previous years." Last is correct - text revised

5. line 77: The term "regionalized" appears odd here. This is a term used in geostatistics for interpolation of point data, but it seems you mean "restricted to certain regions". DONE

6. line 81: While the root cause of arsenic in the IGA system is in the Himalayan sediments, the mechanism are more complicated. I suggest dropping this explanation in order to avoid oversimplification. DONE

7. line 92: Nitrate is sometimes measured as concentration nitrate, and sometimes as concentration nitrate-N. Be specific! No change because not clear in the original report. We followed their usage rather than guess.

---

## Author Comment (AC2) · 20 Oct 2019

We are appreciative of the constructive comments on the paper from all three reviewers. Following here is our detailed response to comments of Reviewer #2. We have considered Reviewer #2's thoughtful comments and made appropriate changes.

Reviewer #2

1. This is an interesting opinion paper on a well-known and significant topic. I enjoyed reading it, especially the review on the case studies and the main problems hampering the effective and sustainable management of groundwater resources. To my best knowledge, the "myth" of groundwater sustainability, and groundwater management in general, belong to many countries, even "advanced" ones, not only Asian.

[Figure]

(response) We appreciate Rev#1's kind comments here. We added the following point that groundwater sustainability belongs to many countries to the Introduction. "For many of these countries and even others outside of Asia, groundwater sustainability is essentially just a myth."

2. The paper is made of two parts: illustration of selected examples and some proposals for a "pragmatic research agenda". The first part is quite good and convincing, although the main conclusions are unfortunately rather obvious and well known nowadays. The collection of cases is not a comprehensive review of groundwater management casein Asia, and it is not meant to be that, but it delivers the message; still, the socio-political conditions are much different among sites such that a comparison is not possible. Perhaps the main focus of the hurdles is on the technical issues, less on the sociopolitical constraints that in many cases lead the process.

(response) We agree with Rev#2 that the focus of the original draft was on technical issues. Yet, as Reviewer #2 indicates, "the socio-political constraints" do indeed lead the process. Given that Rev #1 raised this same issue, we recognize that our "hints" about the importance of this aspect were insufficient.

We addressed this weakness of the paper by adding ∼1 page in the introduction, discussing the socio-economic frameworks, policies. We rewrote Section 3 and added material describing the policy constraints with respect to Pakistan, India and China so it is possible now to compare the status of these countries much more rigorously. The new material (beyond editing what was there) added about $\frac{1}{2}$ page of additional things. You can find this material on lines 194-224 of the revised manuscript.

3. My main reservation is that the exposition looks confusing at times. For instance, the examples continue in Section 3 (by the way, the case of Yemen seems to me quite divorced from the rest standing the particular situation of the area) and one cannot truly see a discontinuity between sections 2 and 3.

(response) We made revisions along the lines suggested by the reviewer to reduce

the confusion. The piece on Yemen is removed as both reviewers suggested. We have retitled Section 2 "2 Trends in Depletion and Contamination of Groundwater Continue to Worsen" and modified the introductory sentence to "In China, India, Pakistan and other hotspots (Figure 1), the impacts to groundwater due depletion and contamination are continuing to worsen for reasons that we will discuss in Section 3." to better differentiate Sections 2 and Section 3. Please note as well the Section 2 has been extensively written, with new material added and other reworked. We think that altogether the changes have added a distinctive focus to Section 2 that differentiates it from Section 3 and reduced the confusion.

4. The lengthy text on the OCWD seems quite out of place and not in line with the rest, which focuses on Asian countries (and do we need Eq.1?). A few sentences would have delivered the same concept. Similar for the Singapore case.

(response) Our rationale with the longer section on OCWD was first to make sure that readers really understood that there are places where quantitatively verifiable groundwater management was taking place. Second we wanted to give a sense of the effort and money needed. This being said, we have trimmed this section substantially and removed the figure. Previously, it was 388 with a figure. Now it is 193 words, no figure and equation 1 removed.

5. The second part, i.e. the delineation of the proposed ideas based on the current management practice in Asia, is much shorter than the first one and not much clear in my view. It definitely needs more elaboration. The Section promises "Groundwater Research Directions" but I can't really find clear and sufficiently elaborated indications. The first item deals with water quality; adding water quality to the management practices seems rather obvious, and it is simplicitly done in several cases, but perhaps I have misunderstood the point (and the short text does not help).

(response) In response to comments from Reviewer #1, we have added two very substantial sections – one to the introduction (1 pg) and the second to section 3 ($\sim$1/2 pg).

The piece in the introduction explains what basic features of a groundwater management scheme should include, and particular socio-economic tools known to incentivize less pumping. In section 3, we have explained how well (or poorly in this case) aligned China, India and Pakistan are with to this framework. This we think is a reasonable response to "more elaboration" comment.

Concerning the promise of "groundwater research directions", the best idea we had was research assuming that a lack of sustainability would create water shortages research could be useful in that area. The present draft has abandoned this idea as considered this too negative, and so research is much less a priority. So we have dropped the promise of research ideas and instead offered a few technology suggestions wireless monitoring and new GRACE and work on traditional approaches. We also recast research in water quality as a first simple step for management as suggested.

6. I agree in principle with the approach of considering the sustainable groundwater management as something that will never materialize, and the derived idea of the worst case scenario. This is something interesting and useful, and sometimes I have seen a similar approach adopted in practical management schemes. However, I see two problems with this approach. First, the analysis of the worst case scenario may anyway need signi?cant resources for data acquisition and the understanding of the groundwater-surface water interactions, and then the several technical problems illustrated in the paper come back again.

Second, the message that may easily come out from this suggestion is the following: forget about management, too dif?cult and expensive, just let things go and prepare for the worst. That would mean the death of the concept of sustainable management and the triumph of "Business As Usual, with likely disastrous consequences on areas characterized by poor or absent management.

(response) Reviewer #2, similar to Dr. Fogg Reviewer #3, is concerned about the negativity in the conclusion that suggested nothing is going to happen with sustainability

and researchers need to get on with adapting to that reality. As mentioned in the comments to Reviewer #3, we have largely rewritten the conclusion. Gone is the negative view that implies business as usual by getting rid the concept of research planning for the worst. We have added more information explaining a new and potentially important role for technology and discussed traditional methods used in India and China in a more positive light. We think these changes have responded to points 1 and 2 in (6).

7. Instead, I think that a less pessimistic alternative would be to provide a management procedure made by subsequent steps of increasing complexity, starting from basic and simple analyses that may guide the management and political decision; in other words, not give up the concept of management. In this perspective, one would rather speak of "feasible management", i.e. based on analyses that can be realistically carried out under the several constraints, starting from the simple concept of safe yield that is relatively easy to estimate in most cases. The governments and stakeholders may start making decision (import food? Invest more on different sources of water? etc.) from those basic and anyway fundamental pieces of information. Role of the scientists and engineers is to try to provide simple rules to stakeholders and managers, while complex management techniques may be affordable only by California or a few other developed regions. To this matter, the list of technical requirements brought by the paper is certainly discouraging. Thus, while the worst case scenario is something worth performing (but how about its uncertainty? Are the future stressors certain?), giving up completely the idea of management might not be so good. Again, I might have misunderstood the concept, and this part of the paper (Section 4) needs further clari?cation and elaboration.

(response) As mentioned in response to point 5, we think in retrospect that promising research directions was an over-reach. So this concept is gone from the previous section title and the paper has been made less pessimistic by discussing new possibilities for future monitoring and management that might come from wireless networks and GRACE. We have encouraged "feasible management" by recasting the MAR approaches they are currently using in a much more positive light.

We did not add specific suggestions about simpler management strategies for two reasons. First, The World Bank has done a great job in promoting practical, country-specific strategies – we added a sentence to the conclusion saying this. As changes prompted by Reviewer #1 has shown, the bottleneck of capacity and fractured policy is so severe that it is difficult to accomplish even simple changes. Second, this is outside the scope of the paper, and adding a small piece to the conclusion we feel would not contribute much.

---

## Author Comment (AC3) · 20 Oct 2019

We are appreciative of the constructive comments and suggestions from all of the reviewers. Following here is our detailed response to comments of Reviewer #3. Dr. Fogg has made excellent points which are considered in our revision.

Reviewer#3 – Dr. Fogg

1. This Opinion paper is a well-written, sobering description of the ongoing crisis of groundwater mismanagement in Asia and prospects for changing course. Despite its negative bottom-line message that the crisis likely cannot be averted, I enjoyed reading the paper and believe the readership will find it interesting and thought provoking.

(response) We appreciate Dr. Fogg's comments here.

[Figure]

2. All of my edits and comments are marked directly in the PDF that is uploaded with this review.

(response) We examined Dr. Fogg's pdf that provided comments on specific phrasing, and more substantive ideas. We made appropriate modifications editorial and otherwise to the manuscript that reflected his suggestion in all instances.

3. My main comment is that the message - that it's highly unlikely for groundwater in Asia to ever be managed sustainably- is too negative. Granted, this is an opinion piece, and the authors are entitled to their opinion, but I think they might be missing an opportunity to provide more impetus for positive change. I worry that the negative message may do more to stifle groundwater management than to produce beneficial change, and all under the assumption that such change is impossible.

(response) Dr. Fogg's comment on the message being too negative is similar to that coming from Reviewer #2. We accepted both their points of view and responding by rewriting the conclusion to brighten it up as well as other parts of the paper. Most importantly we have deleted the idea of "planning for the worst". In other words, we leave the far future to the readers' imaginations without explicitly stating that sustainability won't happen. This has led to changes in the title to section 4 and the concluding paragraph and paragraphs describing research on adaptation. As we mention with specific following comments, we have added several sentences in the conclusion that specifically reflect Dr. Fogg's perspective (below).

Dr. Fogg made a comment before the conclusion that made the point that traditional MAR was a very good thing, rather than some incremental effort that we implied in our initial draft. The revision of the conclusion also put a much more positive "spin" on the MAR efforts in India and China.

4. For added perspective, consider the following:

(a) Any of the needed groundwater information infrastructure would be cheap relative

to the spending these countries are currently doing for construction and maintenance of surface water infrastructure (dams and conveyance). So if they realize they must have something, they can likely find the means to achieve it. One less dam project could free up enough funds for a national groundwater monitoring network. Thailand's Department of Groundwater (yes, there is such a thing) has been doing this nationally since the 1950-60s and hence has been more proactively managing groundwater.

(response) We have no disagreement with what Dr. Fogg has written here. The manuscript was confusing about what kind of infrastructure we were considering to be expensive. We were actually not thinking about informational infrastructure. The infrastructure we wrote about was that required to provide for new sources of water for MAR. Imported water, might, for example, require dams and canals. Using domestic wastewater in India as a source, would require adding, expanding and upgrading sanitary sewer systems, re-imagining the waste-water treatment facilities, MAR and making electric power more reliable. We have added clarification to the text on this point.

(b) The world may be entering a period of change with respect to groundwater management, although it may require considerable coaxing and crises to get there. Since widespread deployment of industrial scale groundwater pumping technologies some 70 yrs ago, very little effort has been devoted to recharging and managing groundwater. In essence, civilization has not yet begun to try to manage groundwater very much, mainly because it has not had to, mainly because of the vastness of most groundwater basin resources. But now that may be starting to change. See the discussion piece: (Trend magazine 2019). I agree - it is questionable whether such change can happen soon enough in Asia, and people should also start preparing for the worst.

(response) We did examine Dr. Fogg's paper in Trend Magazine. The conclusion was modified to reflect this view point through the addition of sentences and reference to Dr. Fogg's paper. This is written in the following section.

(c) One could argue that a big part of the problem is the lack of transparency of groundwater systems, making the state of gw resources easier to ignore. There are technologies coming along that could change this significantly - e.g., low-cost wireless, real-time groundwater level monitoring networks connected to open-source web platforms to track fluctuations in groundwater levels (these may require cellular networks, which are already more extensive in parts of rural Asian than parts of rural America); and future

(response) This point is well taken. We also added words to suggest that monitoring could be a catalyst to increase understanding of the problem "There is, however, some hope that new technologies, may create sufficient visibility as to the severity of the groundwater problems to finally spur action (Fogg, 2019)."

We also add several sentences that made particular reference to real-time monitoring and GRACE follow-on mission as being examples of helpful new technologies.

---

## Author Response (AR1)

We are appreciative of the constructive comments on the paper from all three reviewers. We have considered them carefully and made significant changes to the paper. Most every comment, led to a change in the manuscript that addressed the reviewer's comments. The following pages discuss the specific changes made for each of the reviewers.

**Responses to Reviewer #1:**

The opinion paper by Frank Schwartz and coauthors discusses the lingering groundwater crisis in several Asian countries, some reasons how it could come so far, theoretically feasible technical solutions, and vague research directives. It is clear, that groundwater exploitation is not sustainable in many countries with (semi)-arid climate, including actually large parts of the United States.

1. However, besides climate and land use there are also societal boundary conditions, and these differ tremendously between the countries discussed in the manuscript.

*The original paper draft was focused on mainly technical issues that we considered as hurdles that needed to be overcome for quantitative and verifiable management of large aquifers. Our view was that these issues for many countries in Asia constituted barriers that by themselves would preclude serious efforts towards sustainability. In this respect, the availability of data represents a critical information gap for many countries because you cannot manage something you don't understand. We also used experience from Orange County, California and California more generally to illustrate the true challenges of sustainable management of groundwater from a technical perspective.*

*However, Reviewer #1 quite rightly pointed out that sustainable management also requires a proper legal and socio-economic framework for action. Our paper hinted at the necessity for laws as basis for enforcing limits on withdrawals and synchronization of macroeconomic policies, but the coverage was minimal. Following the reviewer's suggestion have **we expanded the paper significantly** to explain frameworks for action and the various components that contribute to sustainability, and to provide a context for key countries that we considered.*

*We address points 1 and 3 together by adding a long section [lines 35 to 63] in the introduction that describes robust frameworks shown to work in areas of legislation, policy, regulatory/macroeconomic tools. In section "3. What are the Hurdles to Groundwater Sustainability?" we have rewritten and generalized the 2nd "hurdle" concerning data to describe the status of India, Pakistan, and China [lines 189 to 218] with respect to the socio-economic framework discussed in the introduction. The treatment is economical (adding ½ page new) and a significant rewrite of associated material.*

2. The People's Republic of China definitely does lack democratic participation, but it has a long standing tradition of a functional administration, and the economic growth of the last decades has led to the economic foundation for expensive technical solutions, if applicable. We see this in water treatment (both for freshwater and waste water) where tremendous progress has been made in recent years. Of all countries discussed in the manuscript, China is the one where the educational and administrative conditions are the best to implement water-management strategies comparable to those of Southern California - if the Communist Party decides sustainable groundwater management to be an important issue. In contrast, other countries lack the concept of groundwater rights.

*We have added a section explaining the present status of groundwater management in China as well as Pakistan and India. The China piece is part of the longer section described in 1 above (3rd paragraph)*

*lines [189-218] in the manuscript. Assessments by various authors indicate much slower progress in groundwater management than with surface water. We added the sentiment that China would have the financial and technical capacities if the government wished to make progress.*

"However, China does have the fiscal and technical capacity to support projects focused on sustainability."

3. If traditionally the owner of a piece of property is allowed to extract all resources thereof, including groundwater, implementing rules of sustainable groundwater management is doomed to fail. There must be an accepted legal framework stating that you don't own the water of the land that you own, that drilling and operating a new well requires a permit, that the permit can only be issued based on a management plan of the entire resource, that abiding by the rules must be monitored, and that a breach of regulations must be punished. If this basic societal understanding does not exist, sustainability cannot be enforced.

*As mentioned, this is a specific example of the socio-economic issues discussed in 1 and has been addressed in the revision.*

4. I don't think that the authors should put Yemen into the mix of countries to consider. Yemen has been in a Civil War for years, and one cannot expect that anything functions. Almost the same would hold for Afghanistan where the German Geological Survey had spent millions on developing groundwater management rules, including hydrogeological mapping and implementing groundwater monitoring. All of that disappeared when the security of western advisors was no more guaranteed. In such dysfunctional countries, sustainable groundwater management cannot be of high priority. Whereas it could in India.

*Rev#1 and Rev#2 both recommended that we remove this piece. **We have done so**.*

5. The authors present Orange County and Singapore as highly developed regions in which technical solutions for sustainable groundwater management have more or less successfully been implemented, monitored, and maintained. They could add Israel where advanced irrigation techniques and managed aquifer recharge has been developed on a world leading level. Like in Singapore, if even not much more so, Israel is in need of self-sufficiency, has a functional administration, and is home of some of the best engineers worldwide. Hence, when it comes to discussing why sustainable groundwater management appears achievable in Israel but not so much in some of its neighboring countries with similar climate and geology, the societal and governmental boundary conditions must be analyzed to a depth at which geologists and engineers feel uncomfortable. Being a hard-core scientist myself, I lack an in-depth discussion of societal differences among the different countries that can explain differences and give predictions on the chances of implementing sustainable groundwater management practices. Iran, India, China, and Pakistan are quite different countries.

*Israel is certainly worth noting as a country with success in managed aquifer recharge. We have added sentences to the discussion at this point in the paper discussing accomplishments in Israel. In our revision, we have pointed out our view as to why Orange County, Singapore, and Israel have been successful (from paper).*

*"Such sophisticated water management systems are uncommon in Asia. Yet there are several extraordinary examples. The island state of Singapore is home for an innovate collection of management*

*activities creating near self-sufficiency from water imports from Malaysia (Irvine et al., 2014). Drinking and industrial waters come from capturing and treating rainwater captured with urban catchments, the advanced purification of urban wastewater to a product called NEWater, and the addition of desalination plants (Irvine et al., 2014). MAR projects in Israel also provide useful examples. The Dan Region Reclamation Project (also known as Shafdan) uses treated wastewaters from Tel-Aviv and environs for MAR (Cikurel et al., 2012). The system yields 140 $Mm^3/yr$ of high quality water that is pumped 100 km south for irrigation. As of 2012, this was the largest project of its kind in Europe and the Middle East (Cickurel et al., 2012). Israel also depends on the reverse osmosis of seawater with periodic storage of excess water in the Israeli Coastal Aquifer (Ganot et al., 2018).*

*The common characteristics of all three of these successful implementations include (i) extreme shortages of water to the point of exhausting local surface water and groundwater supplies, (ii) technologically advanced and prosperous societies, with modern and reliable infrastructures, and (iii) a manageable problem scope stemming from relatively small populations..”*

*We agree with the reviewers comments in the last few sentences of 5. As mentioned in 1 and 3 we have provided a much improved analysis of the legislative and operational “boundary conditions” to provide a better sense as to which countries are likely to succeed.*

6. The authors rightfully point to water-quality issues related to groundwater management in arid climates and/or regions of intensive agriculture. However, you don't need to go to Asia to realize that salt accumulation in over-exploited aquifers is an issue largely unrecognized by many groundwater managers. In large parts of the western United States, a continuous increase in salinity has been observed in conjunction with declining groundwater levels. At the end of the day, balancing the volume of water is insufficient to obtain sustainability in systems undergoing strong evapotranspiration. We may come to the conclusion that managing the dissolved solids will require more aggressive treatments, such as membrane-based deionization before artificial groundwater enrichment. Luckily, the electricity needed for that can be gained by photovoltaic power in the arid regions that require such treatments the most. Likewise, arsenic (or fluorine) can be removed by technical treatment, but the premise of centralized water treatment is a centralized water supply. In as much, technical solutions for the supply of cities, where centralized treatment options are achievable, must differ from technical solutions for drinking water supply and irrigation agriculture in rural regions. And neither will work without a functional and responsible administration.

*The paper already makes clear there is more to sustainability than taking care of water balances. Indeed this is evident as we mention in both India and Pakistan. We think the points raised here concern remediation membrane-based deionization, arsenic removal, are interesting but are much further in the future. Largely, the character, distribution and concentration remains an informational black hole for all these countries including China. So, we made no changes in this respect but as the reviewer will note (see Reviewer #3 responses) we have addressed technologies as a driver to sustainability.*

7. With respect to research directives, I highly recommend prioritization. Western researchers are interested in exciting science, but that is not always the gateway to practical solutions. Understanding the release and fate of arsenic in deltaic aquifers in south-east Asia is an example of a scientifically challenging question. Alas, among the hundreds to thousands of publications on mechanistic questions related to arsenic in south-east Asia, only a few have been useful to help the people affected. There have been examples in which "cool" science actually contributed to developing sustainable groundwater management strategies, but most of the science is done by the flock of academic sheep following a research bellwether. Most likely, raising the level of education in water-related sciences is the best that university scientists can do to contribute; we need to train people with a solid understanding of hydrogeology and environmental engineering, who hopefully reach positions where they can make decisions. But how a society has to change that responsible decision making by administrative authorities is implemented and accepted, I have no clue.

*We heartily agree with these comments. Obviously, the scope and scale of existing and future problems are too serious to be poking around answering basic-science questions. We have both reworded and added sentences in the conclusion to reflect this view from Rev#1 as follows.*

"There are basic technical approaches that have the potential contribute to sustainability. For example, several countries are already invested in recharge projects, India with their tradition MAR (Davis et al., 2018) and China with their "sponge city" concepts. Significant opportunities exist in identifying strengths and weaknesses with these methods, and in optimizing the benefits for groundwater sustainability. To be most useful, studies should focus on best practices appropriate to the economic and technical capacities of the countries involved."

A few minor comments.

1. line 33: Replace "by right" with "basically". Non-native speakers think you refer to a legal term.

> DONE

2. lines 43-44: Are there only one continuous shallow and one continuous deep aquifer in the entire North China Plain? Otherwise use the plural.  OK as is

3. line 58: Do the percentages refer to India or are the worldwide numbers? The same question refers to the "two prototypical settings for groundwater".   Word "India" added twice in clarification

4. line 63: "recover to the levels of previous years" or "recover from the withdrawals of previous years." Latter is correct.

5. line 77: The term "regionalized" appears odd here. This is a term used in geostatistics for interpolation of point data, but it seems you mean "restricted to certain regions". DONE

6. line 81: While the root cause of arsenic in the IGA system is in the Himalayan sediments, the mechanism are more complicated. I suggest dropping this explanation in order to avoid oversimplification. DONE

7. line 92: Nitrate is sometimes measured as concentration nitrate, and sometimes as concentration nitrate-N. Be specific! No change because not clear in the original report. We followed their usage rather than guess.

**Responses to Reviewer #2**

1. This is an interesting opinion paper on a well-known and significant topic. I enjoyed reading it, especially the review on the case studies and the main problems hampering the effective and sustainable management of groundwater resources. To my best knowledge, the "myth" of groundwater sustainability, and groundwater management in general, belong to many countries, even "advanced" ones, not only Asian.

*We appreciate Rev#1's kind comments here. We added the following point that groundwater sustainability belongs to many countries to the Introduction.* (from paper) *"For many of these countries and even others outside of Asia, groundwater sustainability is essentially just a myth."*

2. The paper is made of two parts: illustration of selected examples and some proposals for a "pragmatic research agenda". The first part is quite good and convincing, although the main conclusions are unfortunately rather obvious and well known nowadays. The collection of cases is not a comprehensive review of groundwater management casein Asia, and it is not meant to be that, but it delivers the message; still, the socio-political conditions are much different among sites such that a comparison is not possible. Perhaps the main focus of the hurdles is on the technical issues, less on the sociopolitical constraints that in many cases lead the process.

*We agree with Rev#2 that the focus of the original draft was on technical issues. Yet, as Reviewer #2 indicates, "the socio-political constraints" do indeed lead the process. Given that Rev #1 raised this same issue, we recognize that our "hints" about the importance of this aspect were insufficient. We addressed this weakness of the paper by adding ~1 page in the introduction, discussing the socio-economic frameworks, policies. We rewrote Section 3 and added material describing the policy constraints with respect to Pakistan, India and China so it is possible now to compare the status of these countries much more rigorously. The new material (beyond editing what was there) added about ½ page of additional things. You can find this material on lines 189-218.*

3. My main reservation is that the exposition looks confusing at times. For instance, the examples continue in Section 3 (by the way, the case of Yemen seems to me quite divorced from the rest standing the particular situation of the area) and one cannot truly see a discontinuity between sections 2 and 3.

*We made revisions along the lines suggested by the reviewer to reduce the confusion. The piece on Yemen is removed as both reviewers suggested.*

*We have retitled Section 2 "2 Trends in Depletion and Contamination of Groundwater Continue to Worsen" and modified the introductory sentence to "In China, India, Pakistan and other hotspots (Figure 1), the impacts to groundwater due depletion and contamination are continuing to worsen for reasons that we will discuss in Section 3." to better differentiate Sections 2 and Section 3.*

4. The lengthy text on the OCWD seems quite out of place and not in line with the rest, which focuses on Asian countries (and do we need Eq.1?). A few sentences would have delivered the same concept. Similar for the Singapore case.

*Our rationale with the longer section on OCWD was first to make sure that readers really understood that there are places where quantitatively verifiable groundwater management was taking place. Second we wanted to give a sense of the effort and money needed. This being said, we have trimmed this*

*section substantially and removed the figure. Previously, it was 388 words with a figure. Now it is 193 words, no figure and equation 1 removed.*

5. The second part, i.e. the delineation of the proposed ideas based on the current management practice in Asia, is much shorter than the first one and not much clear in my view. It definitely needs more elaboration. The Section promises "Groundwater Research Directions" but I can't really find clear and sufficiently elaborated indications. The first item deals with water quality; adding water quality to the management practices seems rather obvious, and it is simplicitly done in several cases, but perhaps I have misunderstood the point (and the short text does not help).

*In response to comments from Reviewer #1, we have added two very substantial sections – one to the introduction (1 pg) and the second to section 3 (~2/3 pg). The piece in the introduction explains what basic features of a groundwater management scheme should include, and particular socio-economic tools known to incentivize less pumping. In section 3, we have explained how well (or poorly in this case) aligned China, India and Pakistan are with to this framework. This we think is a reasonable response to "more elaboration" comment.*

*Concerning the promise of "groundwater research directions", the best idea we had was research assuming that a lack of sustainability would create water shortages research could be useful in that area. The present draft has abandoned this idea as reviewers considered this too negative, and so laying out a research agenda is much less of a priority. So we have dropped the promise of research ideas and instead offered a few technology suggestions wireless monitoring and new GRACE and work on traditional approaches. We also recast research in water quality as a first simple step for management as suggested.*

6. I agree in principle with the approach of considering the sustainable groundwater management as something that will never materialize, and the derived idea of the worst case scenario. This is something interesting and useful, and sometimes I have seen a similar approach adopted in practical management schemes. However, I see two problems with this approach. First, the analysis of the worst case scenario may anyway need significant resources for data acquisition and the understanding of the groundwater-surface water interactions, and then the several technical problems illustrated in the paper come back again.

Second, the message that may easily come out from this suggestion is the following: forget about management, too difficult and expensive, just let things go and prepare for the worst. That would mean the death of the concept of sustainable management and the triumph of "Business As Usual, with likely disastrous consequences on areas characterized by poor or absent management.

*Reviewer #2, similar to Dr. Fogg Reviewer #3, is concerned about the negativity in the conclusion that suggested nothing is going to happen with sustainability and researchers need to get on with adapting to that reality. As mentioned in the comments to Reviewer #3, we have largely rewritten the conclusion. Gone is the negative view that implies business as usual by getting rid the concept of research planning for the worst. We have changed the heading to section 4 and added more information explaining a new and potentially important role for technology. We think these changes have responded to points 1 and 2.*

7. Instead, I think that a less pessimistic alternative would be to provide a management procedure made by subsequent steps of increasing complexity, starting from basic and simple analyses that may guide the management and political decision; in other words, not give up the concept of management. In this perspective, one would rather speak of "feasible management", i.e. based on analyses that can be realistically carried out under the several constraints, starting from the simple concept of safe yield that is relatively easy to estimate in most cases. The governments and stakeholders may start making decision (import food? Invest more on different sources of water? etc.) from those basic and anyway fundamental pieces of information. Role of the scientists and engineers is to try to provide simple rules to stakeholders and managers, while complex management techniques may be affordable only by California or a few other developed regions. To this matter, the list of technical requirements brought by the paper is certainly discouraging. Thus, while the worst case scenario is something worth performing (but how about its uncertainty? Are the future stressors certain?), giving up completely the idea of management might not be so good. Again, I might have misunderstood the concept, and this part of the paper (Section 4) needs further clarification and elaboration.

*As mentioned in response to point 5, we think in retrospect that promising research directions was an over-reach. So this concept is gone from the previous section title and the paper has been made less pessimistic by discussing new possibilities for future monitoring and management that might come from wireless networks and GRACE. We have encouraged "feasible management" by recasting the MAR approaches (both in India and China) are currently using in a much more positive light.*

"There are basic technical approaches that have the potential contribute to sustainability. For example, several countries are already invested in recharge projects, India with their tradition MAR (Davis et al., 2018) and China with their "sponge city" concepts. While, these approaches are represent an important first step to groundwater sustainably, they are no panacea. For example, tradition approaches to water harvesting in India are not well suited for hard-rock areas, impact downstream users, and often lead to more pumping (World Bank, 2010)."

*We did not add specific suggestions about simpler management strategies for two reasons. First, The World Bank has done a great job in promoting practical, country-specific strategies – we added a sentence to the conclusion saying this. As changes prompted by Reviewer #1 has shown, the bottleneck of capacity and fractured policy is so severe that it is difficult to accomplish even simple changes. Second, this is outside the scope of the paper, and adding a small piece to the conclusion would not contribute much.*

**Responses to Reviewer#3 – Dr. Fogg**

This Opinion paper is a well-written, sobering description of the ongoing crisis of groundwater mismanagement in Asia and prospects for changing course. Despite its negative bottom-line message that the crisis likely cannot be averted, I enjoyed reading the paper and believe the readership will find it interesting and thought provoking.

*We appreciate Dr. Fogg's comments here.*

All of my edits and comments are marked directly in the PDF that is uploaded with this review.

*We examined Dr. Fogg's pdf that provided comments on specific phrasing, and more substantive ideas. We made appropriate modifications to the manuscript that reflected his suggestion in all instances.*

My main comment is that the message - that it's highly unlikely for groundwater in Asia to ever be managed sustainably- is too negative. Granted, this is an opinion piece, and the authors are entitled to their opinion, but I think they might be missing an opportunity to provide more impetus for positive change. I worry that the negative message may do more to stifle groundwater management than to produce beneficial change, and all under the assumption that such change is impossible.

*Dr. Fogg's comment on the message being too negative is similar to comments that came from Reviewer #2. We accepted both their points of view and responding by rewriting the conclusion to brighten it up. Most importantly we have deleted the idea of "planning for the worst". In other words, we leave the far future to the readers' imaginations without explicitly stating that sustainability won't happen. This has led to changes in the title to section 4 and the concluding paragraph and paragraphs describing research on adaptation. As we mention with specific following comments, we have added several sentences in the conclusion that specifically reflect Dr. Fogg's perspective (below).*

*Dr. Fogg made a comment before the conclusion that made the point that traditional MAR was a very good thing, rather than some incremental effort that we implied. The revision of the conclusion also put a much more positive "spin" on the efforts in India and China.*

"There are basic technical approaches that have the potential contribute to sustainability. For example, several countries are already invested in recharge projects, India with their tradition MAR (Davis et al., 2018) and China with their "sponge city" concepts."

For added perspective, consider the following:

- Any of the needed groundwater information infrastructure would be cheap relative to the spending these countries are currently doing for construction and maintenance of surface water infrastructure (dams and conveyance). So if they realize they must have something, they can likely find the means to achieve it. One less dam project could free up enough funds for a national groundwater monitoring network. Thailand's Department of Groundwater (yes, there is such a thing) has been doing this nationally since the 1950-60s and hence has been more proactively managing groundwater.

*We have no disagreement with what Dr. Fogg has written here. The manuscript was confusing about what kind of infrastructure we were considering to be expensive. The infrastructure we wrote about was that required to provide for new sources of water for MAR. For example, imported water, might require dams and canals. Using domestic wastewater in India as a source, would require adding, expanding and*

*upgrading sanitary sewer systems, re-imagining the waste-water treatment facilities, MAR and making electric power more reliable. We have added clarification to the text on this point.*

- The world may be entering a period of change with respect to groundwater man-agement, although it may require considerable coaxing and crises to get there. Since widespread deployment of industrial scale groundwater pumping technologies some 70 yrs ago, very little effort has been devoted to recharging and managing ground-water. In essence, civilization has not yet begun to try to manage groundwater very much, mainly because it has not had to, mainly because of the vastness of most groundwater basin resources. But now that may be starting to change. See the discussion piece: https://trend.pewtrusts.org/en/archive/spring-2019/groundwater-the-resource-we-cant-see-but-increasingly-rely-upon. I agree - it is questionable whether such change can happen soon enough in Asia, and people should also start preparing for the worst.

*The conclusion was modified to reflect this viewpoint through the addition of a sentence and reference to Dr. Fogg's paper.*

(from paper) "It may also be that groundwater is so plentiful that it has never been a concern (Fogg, 2019)."

- One could argue that a big part of the problem is the lack of transparency of groundwater systems, making the state of gw resources easier to ignore. There are technologies coming along that could change this significantly - e.g., low-cost wireless, real-time groundwater level monitoring networks connected to open-source web platforms to track fluctuations in groundwater levels (these may require cellular networks, which are already more extensive in parts of rural Asian than parts of rural America); and future

*We also added words to suggest that monitoring could be a catalyst to increase visibility.*

"There is, however, some hope that new technologies may create sufficient visibility as to the severity of the groundwater problems to finally spur action (Fogg, 2019)."

*We also made particular reference to real-time monitoring and GRACE follow-on mission as being examples of helpful new technologies.*

[revised manuscript text omitted]

---

## Author Response (AR2)

The Editor requested that we review the manuscript to assure technical requirements etc. were met.

We carefully reviewed the manuscript and corrected a few errors with respect to citations. We made just a few grammatical edits. We think we have responded according to the editor's comments